# Multi-omics profiling reveals atypical sugar utilization and a key membrane composition regulator in *Streptococcus pneumoniae*

Vincent de Bakker [1,2], Xue Liu [1,3], Jonah Tang[4], Matthew Barbisan[4], Jonathon L. Baker [4] & Jan-Willem Veening [1] ✉

The human body comprises many different microenvironments, each with their own challenges for microorganisms to overcome in order to survive and possibly cause infection. The human pathogen *Streptococcus pneumoniae* is notoriously flexible in this regard, and can adapt to a wide range of host niches, including the nasopharynx, lungs, and cerebrospinal fluid. However, the molecular and genetic foundation of this ability remain largely uncharted. In this work, we demonstrate that niche adaptation imposes genome-wide changes on multiple levels, including gene essentiality, expression and membrane lipid composition, by using infection-mimicking growth conditions. In general, we show that gene expression and fitness profiling couple orthogonal sets of genes to environmental stimuli. For instance, import (*manLMN*) and catabolism (*nagAB*) genes are required, but not differentially expressed during growth on N-acetylglucosamine (GlcNAc), opposite to the pattern of other amino sugar metabolism pathways. Surprisingly, we find that pneumococci do not necessarily prefer glucose over GlcNAc and that uptake of GlcNAc in absence of subsequent catabolism is toxic. Moreover, we identify a previously overlooked fatty acid saturation regulator, FasR, controlling membrane composition, rendering it important during heat stress. As nutrient availability and temperature fluctuations are distinctive facets of infection environments, these findings may inform anti-infective strategies.

The human body is the natural habitat of many microorganisms. Various parts of the body impose distinct environmental conditions, such as differences in nutrient availability, surrounding temperature, or host immune responses. Nevertheless, some organisms are highly adaptable and manage to survive and grow in multiple human niches. Among these is the bacterium *Streptococcus pneumoniae* (the pneumococcus); a commensal of the human nasopharynx that can cause severe disease states upon invasion of other niches, such as pneumonia in the lungs, or meningitis in the cerebrospinal fluid[1–3]. As such,

pneumococci are the main cause of lower respiratory tract infections worldwide, and are associated with most deaths in children under five years old[4,5].

This high degree of niche flexibility is genomically well reflected by the large number of carbohydrates the pneumococcus can use as a carbon source[6]. Indeed, this obligate fermenter has an arsenal of 28 transport systems to consume at least 32 different carbon substrates[7]. Many of these systems display a two-way redundancy: substrates can often enter the cell through multiple importers, and many transporters

[1]Department of Fundamental Microbiology, Faculty of Biology and Medicine, University of Lausanne, Lausanne, Switzerland. [2]Department of Microbiology, Harvard Medical School, Boston, MA, USA. [3]Department of Pathogen Biology, Base for International Science and Technology Cooperation: Carson Cancer Stem Cell Vaccines R&D Center, International Cancer Center, Shenzhen University Health Science Center, Shenzhen, China. [4]Department of Biomaterial & Biomedical Sciences, School of Dentistry, Oregon Health & Science University, Portland, OR, USA. ✉e-mail: Jan-Willem.Veening@unil.ch

can import multiple substrates[7]. Of these transporters, 21 are phosphoenolpyruvate–carbohydrate phosphotransferase systems (PTS), which canonically also function in carbon catabolite control (CCR): the classic process through which bacteria shut down the import and metabolism of other carbon sources in the presence of a preferred one, conventionally glucose[8,9]. Indeed, pneumococci readily take up glucose when provided while growing on galactose, mannose, or N-acetylglucosamine (GlcNAc)[10]. Of these sugars, galactose is consumed last, and while GlcNAc is used first, mannose can be taken up simultaneously[11]. Mannose and GlcNAc are both taken up through the PTS ManLMN, which also imports glucose and appears to be a key component of pneumococcal CCR[7,12]. Central to this process is the catabolite control protein A (CcpA), which interacts with a PTS component to transcriptionally regulate carbon metabolism genes and has been shown to affect transcript levels of up to 19% of pneumococcal genes[13]. Given this dominant regulatory role, it is unsurprising that *ccpA* and *manLMN* are important for virulence, highlighting the importance of metabolic regulatory systems to maintain fitness in changing host niches[14–17].

This is also true for lipid metabolism, where FabT controls expression of the type II fatty acid synthesis (FASII) genes in *S. pneumoniae*[18]. An important exception is *fabM*, required for the production of unsaturated fatty acids, whose regulation remains obscure[19,20]. On top of de novo biosynthesis, pneumococci are also capable of exogenous (e.g., host-derived) fatty acid acquisition through the FakAB system[21–23]. Whether synthesized or imported, fatty acids differ in acyl chain length and saturation degree, and can be incorporated into the plasma membrane as key building blocks of phospholipids[24]. So, the bacteria can modulate their membrane composition by regulating relative intracellular availability of different fatty acid types, in turn affecting membrane properties such as viscosity or thickness[25]. Since these properties are also affected by environmental factors, like temperature, the regulation of fatty acid metabolism to maintain membrane homeostasis is a crucial niche adaptation strategy[26]. Indeed, the textbook example of this phenomenon is homeoviscous adaptation, in which cells adjust the saturated to unsaturated fatty acid (SFA:UFA) ratio in the membrane to counteract temperature-mediated changes in fluidity[27].

Given the hyperadaptive lifestyle of *S. pneumoniae*, a significant part of its genome may be expected to be dedicated to such niche adaptation mechanisms. However, many pneumococcal genes remain of unknown function (as much as 14% of all genes for strain D39V, NCBI accession CP027540[28,29]), in part exacerbated by its extensive pangenome[30]. We have previously examined transcriptomes in several infection-relevant growth conditions to infer gene function and probe genotype-phenotype links at scale[31]. Although insightful, we note that transcriptional changes are generally only moderately reflected on the protein level[32–35], and are not necessarily informative on gene essentiality[36–41]. The latter has previously been quantified on a genome-wide level using Tn-seq and CRISPRi-seq approaches, providing gene function insights as well[42,43]. However, genome-wide Tn-seq coupled to RNA-seq studies in *S. pneumoniae* indeed showed poor links between transcriptional stress responses and gene essentiality[36].

Here, we address these shortcomings by measuring gene expression on both the transcriptomic and proteomic levels, and by interrogating both fitness loss and gain effects of nearly all *S. pneumoniae* D39V genes using CRISPRi-seq[44]. Since this includes baseline essential genes (in contrast to Tn-seq), it allows us to draw comprehensive comparisons between expression levels of all genes and their impact on the bacterium's fitness, and link these to distinct growth conditions. In this way, we show that niche adaptation happens on all these regulatory levels, and that expression and fitness indeed provide orthogonal, complementary sets of information on the cell state. Specifically, we show that these data can shed light on concrete molecular processes by delving into an unexpected requirement for

N-acetylglucosamine catabolism, even in the presence of glucose, and a hitherto overlooked putative membrane composition regulator. Together, these results provide a systems-level perspective of pneumococcal niche adaptation and lay the groundwork for extensive gene function studies.

## Results

### Infection-relevant conditions impose distinct fitness landscapes

Previously, we described pneumococcal transcriptomic profiles for a set of 22 infection-relevant growth conditions, based on chemically defined media[31]. Here, we set out to assess how these differences in expression relate to the importance of those genes for growth in each environment. To this end, we generated genome-wide fitness profiles using CRISPRi-seq with an IPTG-inducible dCas9-sgRNA library in a subset of growth conditions that previously showed clearly distinct transcriptomes[43].

The selected conditions aim to simulate the host environment as encountered by pneumococci in the nasopharynx (as in colonization), lungs (pneumonia), blood (bacteremia) or cerebrospinal fluid (meningitis), mostly in terms of temperature, acidity, and nutrient availability (Supplementary Table 1)[31]. We also included the commonly used complex media THY and C+Y, the latter both with and without chemically inducing competence by the addition of synthetic competence-stimulating peptide 1 (CSP-1), whose transcriptional responses have been well characterized[45–47].

We observed clear condition-specific fitness landscapes among all conditions tested (Fig. 1a), with 362 operons (corresponding to 615 genes) causing conditional fitness effects ($\Delta|\log_2 FC| > 1$, $P_{adj} < 0.05$, Fig. 1b), and a core essentialome of 139 operons (259 genes) ($\log_2 FC < 1$, $P_{adj} < 0.05$, Supplementary Fig. 1a, Supplementary Data 1). We made these essentiality calls easily accessible through our recently updated online genome browser PneumoBrowse 2 (https://veeninglab.com/pneumobrowse)[29]. Apart from the CRISPRi induction effect itself (PC1), the largest source of fitness variation was associated with the difference between complex and defined media (PC2, Fig. 1a). Indeed, most of the genes driving this distinction had metabolic functions, such as members of the Shikimate pathway, enabling aromatic amino acids biosynthesis, and the AmiACDEF oligopeptide permease system (Fig. 1b). Specifically, the former were exclusively essential in the chemically defined media, whereas the latter showed the opposite profile, suggesting that bacteria scavenge these amino acids in the form of peptides under nutrient-rich conditions, as previously reported[48]. These patterns were not surprising, since nutrient availability constitutes the biggest difference between the growth conditions used here (Supplementary Table 1). In particular, the presence of one key nutrient appeared to cause clear, conditional growth phenotypes: the amino sugar N-acetylglucosamine (GlcNAc, Fig. 1b). We next sought to validate the corresponding gene hits.

### GlcNac catabolism is essential upon import despite glucose availability

The genes *nagA* and *nagB* encode the two proteins responsible for the breakdown of GlcNAc, feeding into glycolysis via ᴅ-fructose 6-phosphate (Fig. 1b)[11,49]. Accordingly, both genes were exclusively essential in the nasopharynx- and lung-mimicking conditions (NMC/LMC), where this was the only sugar added to the medium (Fig. 1b, Supplementary Table 1). To test if we could indeed change the fitness status of these genes from dispensable to essential in C + Y as well, we replaced their regular sugars, glucose and sucrose, with GlcNAc. Although we could confirm the effect even in this complex medium with growth curves of knockout and aTc-inducible complementation strains, we noticed these knockout strains also displayed a severe growth defect on an equimolar mix of both GlcNAc and glucose (Fig. 2a). This growth defect was reproducible and dependent on GlcNAc concentration (Supplementary Fig. 1b). This result was

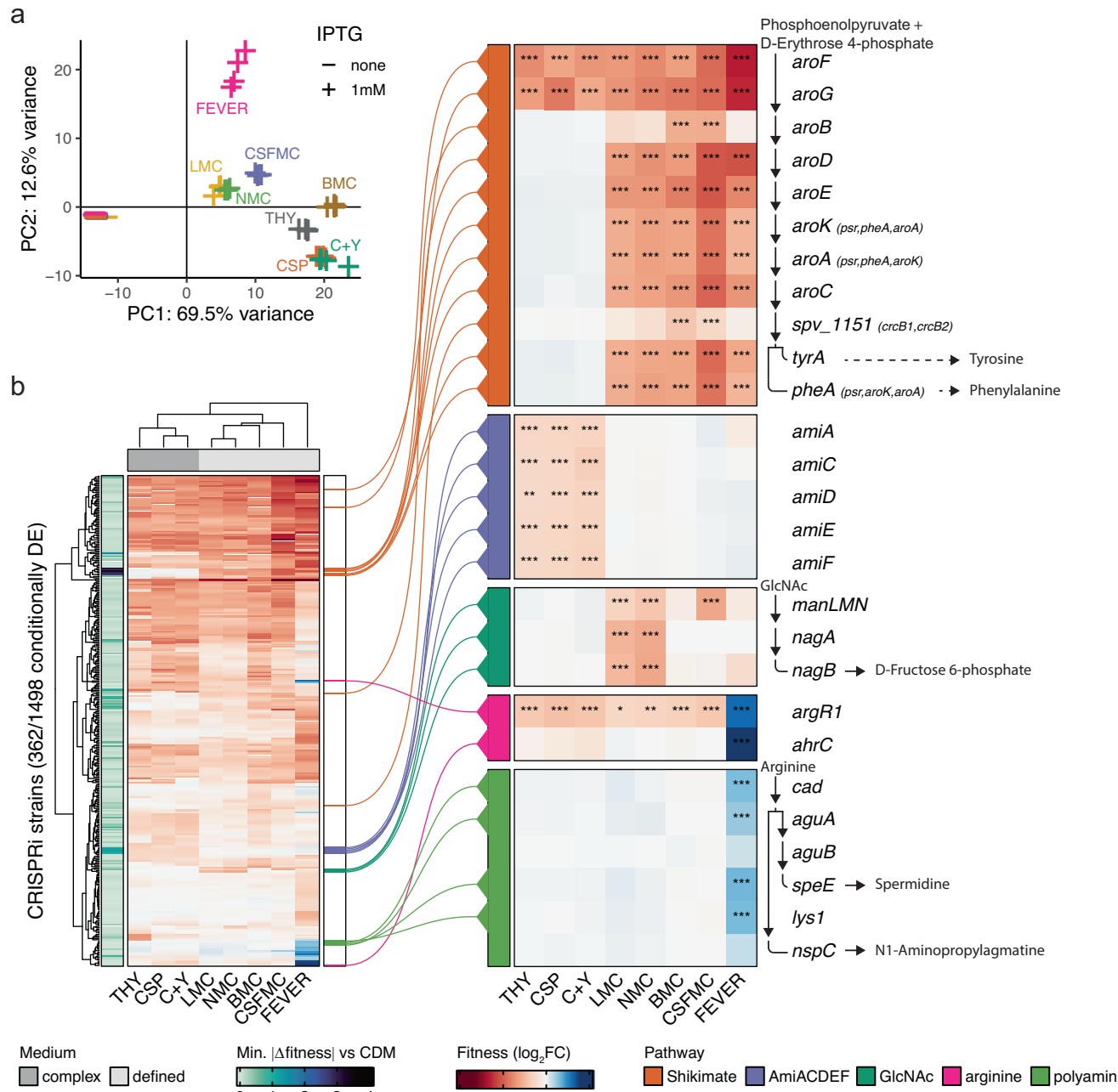

**Fig. 1 | Genome-wide fitness profiles by CRISPRi-seq. a** Principal component (PC) analysis demonstrating heterogeneity in CRISPRi strain composition was reproducibly driven by (1) CRISPRi induction following isopropyl β-ᴅ-1-thiogalactopyranoside (IPTG) addition and (2) medium complexity. Data were normalized using the rlog transformation implemented in DESeq2[78]. **b** Overview of all CRISPRi strains that were differentially enriched (DE) between at least two growth conditions (Δ| $\log_2 FC| > 1$, $P_{adj} < 0.05$). For each strain, the minimum absolute differential fitness score between any complex versus any chemically defined medium (CDM) condition is also shown, highlighting the strongest consistently differentially essential genes between these medium types. Inset heatmaps show major metabolic pathways discussed in this work, where rows are ordered according to the respective pathway, with arrows indicating enzymatic reactions displayed alongside the corresponding gene names. Genes occurring in the same operon were targeted by the same sgRNA, as indicated in brackets, and as a result, the fitness scores for *aroK*, *aroA*, and *pheA* are identical in the inset heatmap. *spv_1151* encodes chorismate mutase[48]. Intermediate products of *tyrA* and *pheA* are transaminated to yield tyrosine and phenylalanine. GlcNAc N-acetylglucosamine, THY Todd–Hewitt broth with yeast extract, CSP competence-stimulating peptide, C+Y C-medium with yeast extract, L/N/B/CSFMC lung/nasopharynx/blood/cerebrospinal fluid-mimicking condition, FEVER CSFMC at 40 °C; see also Supplementary Table 1. Differential enrichment was tested using DESeq2 (negative binomial generalized linear model with a Wald test) with two-tailed tests and *P*-values adjusted for false discovery rate[78]. Source data are provided in Supplementary Data 1.

surprising, because glucose is commonly believed to be the preferred carbon source in *S. pneumoniae*, which boasts a classic CcpA-based carbon catabolite repression system[10,13]. In addition, neither *nagA* nor *nagB* has any known role in glucose uptake or catabolism. So, in the presence of sufficiently high glucose levels, as in this 1:1 molar sugar mix, we would expect all strains to grow well, including the mutants.

We therefore decided to measure changes in the glucose concentration in the media during exponential growth using glucose oxidase assays, as a proxy for glucose uptake. Whereas all strains

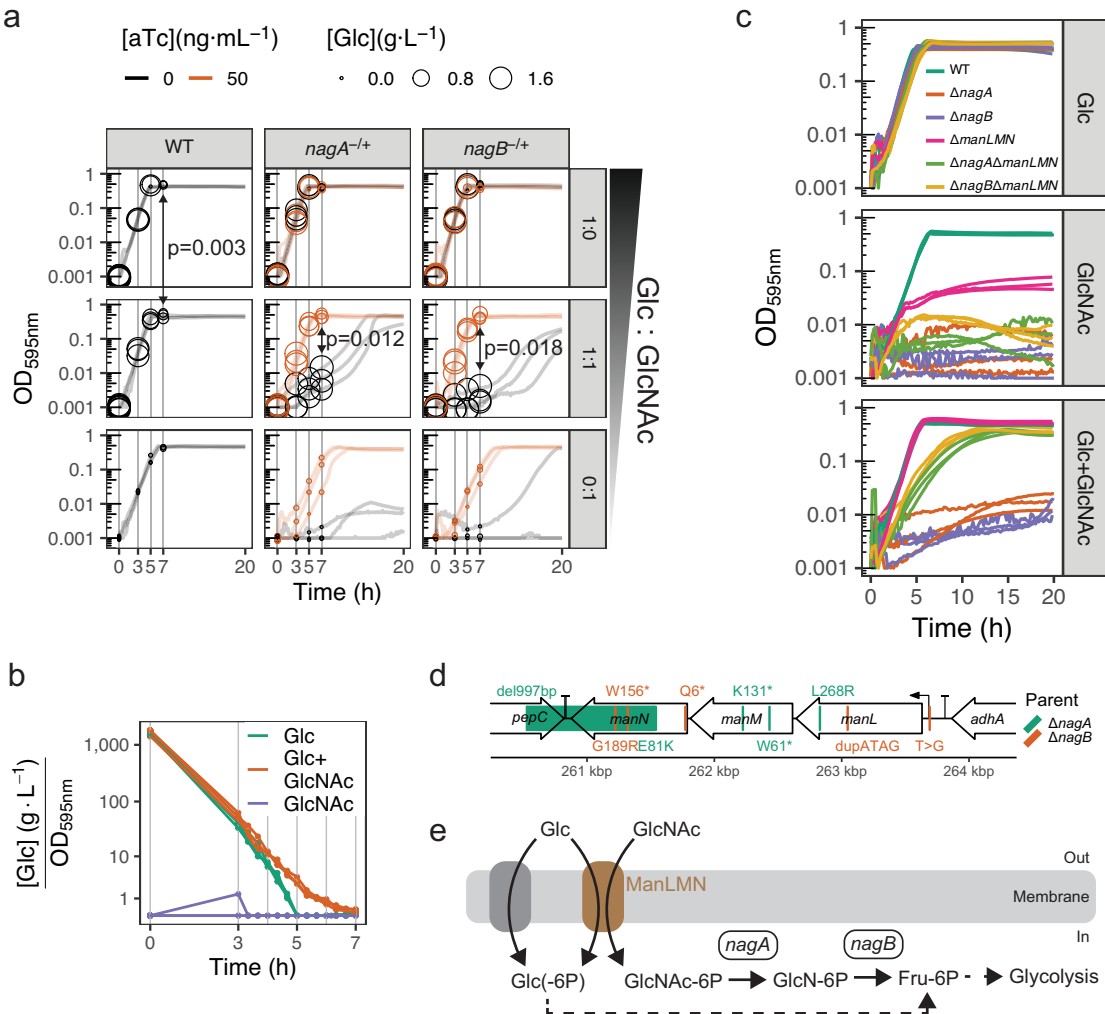

**Fig. 2 | N-acetylglucosamine (GlcNAc) catabolism genes *nagA* and *nagB* are essential on a sugar mix with glucose (Glc). a** Growth curves of the wild-type (WT) strain as well as anhydrotetracycline (aTc)-inducible ectopic complementation strains $P_{tet}$-*nagA* and $P_{tet}$-*nagB* in which the native genes are deleted, respectively (*nagA*$^{-/+}$, *nagB*$^{-/+}$). Glucose concentrations in the medium are displayed as rings, superimposed on the curves of the replicate in which they were measured. Strains were grown with or without aTc on medium with either glucose, GlcNAc, or both in equimolar concentrations. Glucose concentration differences were assessed by paired, two-tailed *t*-tests to account for nesting within plates, and corresponding *P*-values are shown. **b** Medium glucose concentration normalized by OD for WT. Separate curves for glucose concentration and OD can be found in Supplementary Fig. 1c. **c** Growth curves for WT, single, and double mutants on medium containing either glucose, GlcNAc, or both in equimolar concentrations. **d** Schematic of the *manLMN* locus showing mutations found in *nagA/B* suppressor mutants. del: deletion, *: stop codon, dup duplication. **e** Deletion of either *nagA* or *nagB* could result in the accumulation of intermediate sugar-phosphate metabolites when grown on GlcNAc, even in the presence of glucose. This accumulation could then be alleviated by also deleting *manLMN*, as there are alternative import routes for glucose. Although the identity of these alternative glucose transporters has not yet been established, we and others have shown that *manLMN* by itself is not essential to support growth on glucose (**c**), pointing indeed to the presence of functionally redundant sets of transporters[7,12]. Biological triplicates are shown for all experiments. Source data are provided as a Source Data file.

readily depleted the medium of all glucose when no GlcNAc was added, the Δ*nagA* and Δ*nagB* mutants failed to do so in the same time window when both sugars were present (Fig. 2a). Moreover, even in wild-type (WT) bacteria glucose uptake was slower when grown on the sugar mix compared to the glucose-only medium, suggesting GlcNAc presence inhibits glucose import (Fig. 2a). We validated this result by repeating the assay measuring every 20 min instead of every 2 h, and found a consistent 2-h delay in complete glucose depletion in the presence of GlcNAc (Fig. 2b). Despite this considerable slowdown in glucose uptake, supplementation with GlcNAc did not impact the WT growth rate (Supplementary Fig. 1c). This implies that the bacteria take up GlcNAc even in the presence of glucose, making up for the apparent reduced sugar import to fuel growth. These results suggest that GlcNAc competes for import with glucose, which might therefore not necessarily be the preferred carbon source for *S. pneumoniae*.

Since GlcNAc versus glucose uptake seemed competitive, and Δ*nagA*/Δ*nagB* mutants did not appear to import glucose, we hypothesized that their growth defect on the sugar mix was likely caused by the inability to break down GlcNAc after import. Limiting intracellular GlcNAc concentrations by knocking out its importer should then alleviate the growth defect. The main GlcNAc importer has previously been reported to be the phosphotransferase system ManLMN, which was indeed conditionally, albeit not exclusively, essential in NMC and LMC in our CRISPRi-seq assay (Fig. 1b)[7,50]. Additionally, *manLMN* deletion caused a strong fitness defect on a GlcNAc-only medium, further confirming its importance in GlcNAc consumption (Fig. 2c). However, on the sugar mix, *manLMN* deletion drastically increased Δ*nagA* and Δ*nagB* viability, in line with our hypothesis (Fig. 2c). Moreover, we performed a suppressor screen by plating both Δ*nagA* and Δ*nagB* mutants on agar containing both sugars. We isolated 20

colonies, 10 for each parent strain, all of which phenocopied the partial rescue of the double knockout strains in liquid (Supplementary Fig. 1d). Sequencing results of 10 of these isolates, five for each parent strain, revealed that all of them were mutated in the *manLMN* locus, indeed suggesting selection for GlcNAc import reduction (Fig. 2d).

Taken together, our results reveal an unusual sugar preference in *S. pneumoniae* and imply the presence of toxic intermediates in the GlcNAc catabolic pathway. Since all intermediates are sugar phosphates (Fig. 2e), and the toxicity is seen in both the Δ*nagA* and Δ*nagB* mutants (Fig. 2a, c), we speculate this could be a form of sugar-phosphate stress, as has been described for other species and sugars[51].

### Heat stress imposes specific genetic requirements

Although metabolic effects dominated the conditional fitness profiles by separating complex media from defined ones, the FEVER condition appeared more distinct (Fig. 1a, b). Since bacteria in this condition were grown at an elevated temperature of 40 °C, we sought to explore whether this could explain FEVER-specific effects. To this end, we compared it with growth in CSFMC: the same medium, but at 37 °C (Supplementary Table 1). Although growth was approximately three times slower at 40 °C compared to 37 °C, we ensured all of these samples were grown for the same pooled average of seven generations to allow the CRISPRi strains to compete for an overall equal number of cell divisions (Supplementary Table 1, Supplementary Data 1).

Strikingly, we uniquely observed fitness gain effects for the knockdown of certain genes in FEVER (Fig. 1b). Among the strongest of these was arginine metabolism regulator *argR1* (Figs. 1b and 3a). ArgR1 functions as a dimer with AhrC, whose gene indeed had a similar effect on fitness[52]. Since these two genes are located more than 780 kbp apart, this cannot be due to CRISPRi polar effects. Most other strong fitness gains were achieved through repression of genes neighboring *ahrC* (*xseA, xseB, ipsA-spv_1064, recN*), which we therefore expect to be due to polarity (Fig. 3a, Supplementary Data 1). In addition, arginine is the precursor for polyamine biosynthesis in *S. pneumoniae*, and most of the genes involved in this pathway also showed mild fitness gain effects upon repression, of which four significantly so (log$_2$FC > 1, $P_{adj}$ < 0.05, Figs. 1b and 3a, Supplementary Data 1)[53]. Knockdown of these genes thus likely increases intracellular arginine levels, an effect also associated with *ahrC* and *argR1* repression[52]. Together, these results suggest arginine retention is favorable for *S. pneumoniae* at higher temperatures, although this remains to be investigated more deeply.

As expected, repression of genes encoding known heat-shock proteins such as *groEL/ES, ftsH*, and the *hrcA-grpE-dnaK-spv_2171-dnaJ* operon resulted in strong fitness losses (Fig. 3a, Supplementary Data 1)[54]. The strongest FEVER-specific fitness loss was, however, caused by a two-gene operon hitherto not associated with heat stress: *comEB-spv_0647* (Fig. 3a). We first validated this result with an operon-based deletion and complementation strain, and subsequently confirmed that deletion of *spv_0647*, but not *comEB*, causes growth defects at higher temperatures (Supplementary Fig. 2a).

### SPV_0647 is critical for membrane homeostasis at high temperatures

Since *spv_0647* encodes a hypothetical, putative transcriptional regulator (TetR family), we made a clean knockout strain and performed RNA-seq comparing its transcriptome to that of WT at two different temperatures, in order to uncover its potential regulon (Supplementary Data 2). To limit toxic side effects, and because the growth defect was already visible at 37 °C, we opted to draw the comparison between this temperature and 30 °C, where the growth defect of the mutant was virtually absent (Supplementary Fig. 2a).

Indeed, the transcriptomes diverged substantially at 37 °C, confirming a temperature-dependent effect (Fig. 3b). While many more

genes were differentially expressed at 37 °C, *spv_0647* mRNA was always depleted in the mutant regardless of temperature, as expected (Fig. 3c). Similarly, multiple genes involved in fatty acid metabolism were also consistently downregulated, most notably *fabM, fakB3*, and fatty acid biosynthesis cofactor biotin transporter *bioY* (Fig. 3c).

We argued that dysregulation of fatty acid metabolism could alter membrane composition, affecting properties such as its permeability and fluidity. As temperature changes themselves are also known to affect these properties, we hypothesized that the growth defect of the mutant at higher temperatures is the result of an interaction between heat stress and fatty acid dysregulation, compromising membrane integrity. In turn, membrane weakening would likely disrupt many downstream processes, especially those involving transmembrane transport or membrane-anchored proteins. Changes in, for instance, protein localization or in- and efflux of nutrients and signaling molecules could stimulate or inhibit sensing pathways, potentially offering an explanation for the massive, global transcriptome divergence we observed at 37 °C, including pyrimidine biosynthesis (PyrR), bacteriocin production (Blp), and multiple sugar metabolism loci (maltosaccharide, cellobiose, beta-glucoside; Fig. 3c).

Fatty acid biosynthesis in *S. pneumoniae* is controlled by FabT, regulating expression of the FASII operon[18]. However, the first gene of the locus, *fabM*, is not controlled by FabT[19]. FabM balances substrate availability for the production of unsaturated versus saturated fatty acids, and a mutant cannot synthesize unsaturated variants[20,55]. Although pneumococci can only make mono-unsaturated fatty acids, they are able to take up exogenous poly-unsaturated fatty acids through binding of FakB3 and subsequent phosphorylation by FakA, making these species also available for membrane incorporation[21,23,56–58]. As such, downregulation of either *fabM* or *fakB3* should yield relatively higher saturated to unsaturated fatty acid ratios (SFA:UFA) in the membrane, which is indeed known to be a major factor affecting membrane properties like fluidity[26,27]. Moreover, *fakB3* is located directly upstream of *spv_0647* on the chromosome, in antisense orientation (Fig. 3d). Since the genes encoding transcriptional regulators are often situated adjacent to the genes they control, it is tempting to speculate on a potential *fakB3*-regulating role for SPV_0647. Of note, transcriptional repression of neither *fakA/B3* nor *fabM* resulted in significantly decreased fitness at 40 °C (Fig. 3a), implying these genes are not individually responsible for the heat-sensitive phenotype.

To get more clues regarding SPV_0647 function, we compared its predicted folding to determined crystal structures in the RCSB Protein Data Bank (PDB) using FoldSeek[59]. Strikingly, we found that despite strong sequence dissimilarities, the protein is predicted to have a structure similar to multiple known TetR-like lipid metabolism transcriptional regulators, including activators (Table 1, Supplementary Fig. 2b, Supplementary Data 3). These hits point to a role of SPV_0647 in fatty acid regulation.

We next revisited our recently published *S. pneumoniae* D39V genome-wide genetic interaction data generated by dual CRISPRi-seq[60], and found negative interactions of the *spv_0647-comEB* operon with *fakA* and *fabM* (Fig. 3e). Strikingly, this effect was not observed for the other two sgRNAs targeting the directly downstream *fabT* and *fabK* in the FASII locus. This implies the interaction only involves *fabM*. Furthermore, *asp23* displayed a similar negative interaction, which is likely a polar effect, given its location directly upstream of *fakA* without intermediate terminator[28]. The negative interaction with *spv_1145*, a dNTP triphosphohydrolase, is likely due to the polar knockdown of *comEB*, a dCMP deaminase. Lastly, the genetic interaction with *spv_1294* implies a potential role for the encoded hypothetical protein in either DNA or fatty acid metabolism. These results suggest *spv_0647* might have a function in FakA- and FabM-related processes, influencing fatty acid saturation levels in the membrane.

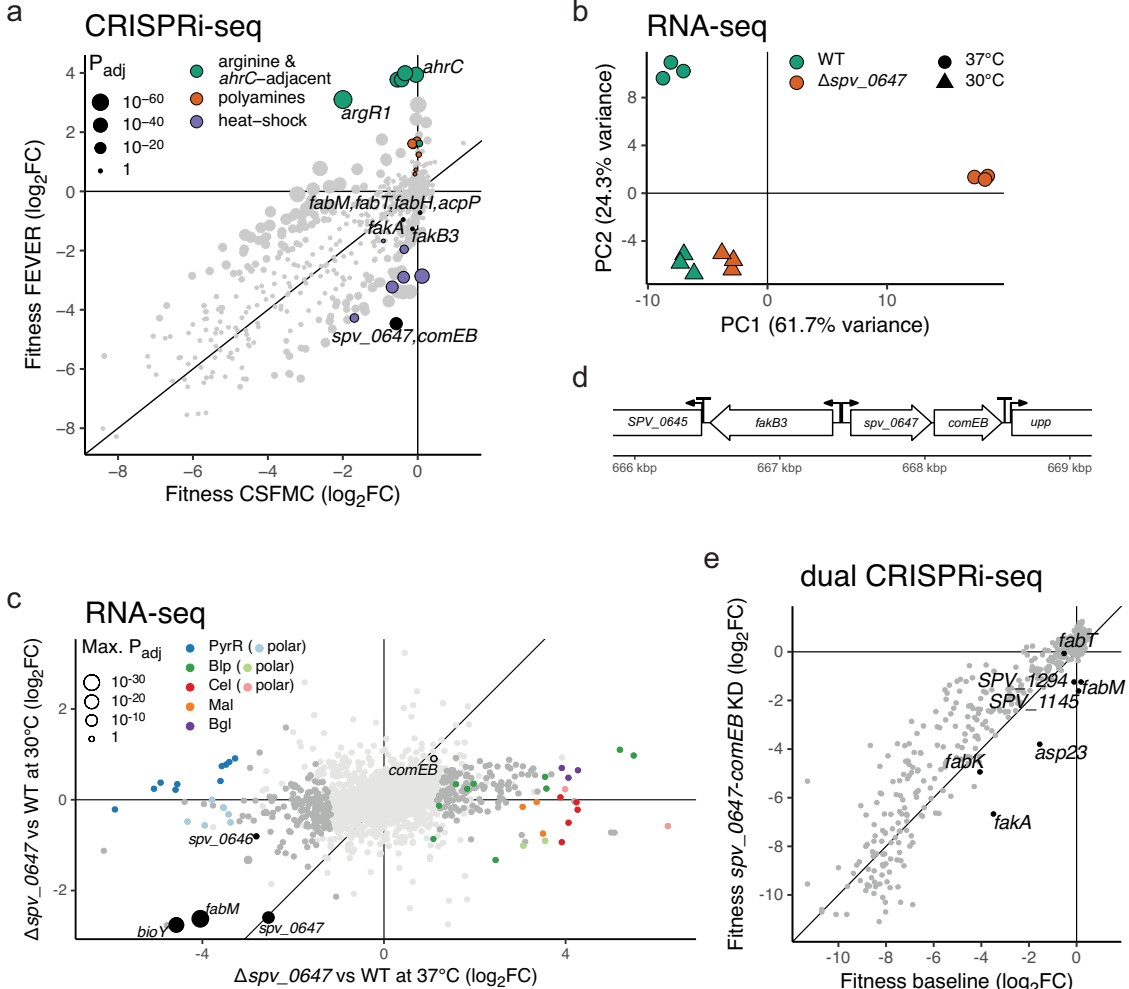

**Fig. 3 | *spv_0647* is important during heat stress and functionally related to genes modulating fatty acid saturation levels. a** Fitness in FEVER condition (40 °C) compared to CSFMC (37 °C) as measured by CRISPRi-seq. Adjusted *P*-values represent significance testing for a fitness difference of at least one log₂FC. Arginine and polyamine pathway genes correspond to those in Fig. 1b. The highlighted *ahrC*-adjacent genes are its chromosomal neighbors *xseA*, *xseB*, the *ipsA-spv_1064* operon, and *recN*. Labeled heat-shock genes represent *ftsH*, *clpP*, *clpE*, and the *ctsR-clpC*, *groES-groEL*, and *hrcA-grpE-dnaK-spv_2171-dnaJ* operons. Genes targeted by the same sgRNA due to operon structure are labeled as separated by commas. **b** Principal component analysis of Δ*spv_0647* mutant and WT transcriptomes measured by RNA-seq at either 30 or 37 °C; based on rlog normalization[78]. **c** Differential expression of genes between the Δ*spv_0647* mutant and WT strains at either 37 or 30 °C. The maximum adjusted *P*-value representing differential expression at either temperature is shown per gene. Genes not differentially expressed at either temperature (*P*adj > 0.05 for |log₂FC| > 1) are displayed in light

gray. Significant hits with high fold changes were grouped and colored by pathway according to annotations in PneumoBrowse 2 and literature (PyrR, pyrimidine metabolism: *pyrBDdEFKR*, *spv_2513*, *carAB*, *uraA*; Blp, bacteriocin production: *blpABCHKRSTYZ*; Cel, cellobiose uptake: *celABCDR*, Mal, maltosaccharide metabolism: *malPQ*, *spv_2SS2*, Bgl, beta-glucoside metabolism: *bglGFA-2*), where lighter hues indicate presumed polar effects of directly neighboring genes (PyrR: *spv_0610-0614*, *lytB*; Blp: *pncP*, *spv_2495*, Cel: *spv_0278*, *spv_0282*)[7,28,96–98]. **d** Genomic neighborhood of *spv_0647*, including promoters and terminators. **e** Genetic interactions of *spv_0647* according to dual CRISPRi-seq data of Dénéréaz et al.[60]. Shown are the fitness effects of each sgRNA in the absence (baseline) or presence of the sgRNA targeting the *spv_0647-comEB* operon. KD knockdown. Differential enrichment of both sgRNAs and transcripts was tested using DESeq2 (negative binomial generalized linear model with a Wald test) with two-tailed tests and *P*-values adjusted for false discovery rate[78]. Source data are provided in Supplementary Data 1, 2 and as a Source Data file.

## FasR (SPV_0647) controls fatty acid saturation balance mediated by FabM

To test whether the growth phenotype was brought about by *fabM* or *fakB3* downregulation (Fig. 3c), we tried to rescue the bacteria by overexpression of either or both genes in the Δ*spv_0647* mutant. *fakB3* overexpression improved growth marginally at best, which was not surprising, as the growth medium did not contain an explicit excess of polyunsaturated acids. However, *fabM* overexpression clearly restored growth (Fig. 4a). This implies that generating a surplus of (substrate for) unsaturated fatty acids rescues the mutant, which might thus lack sufficient levels of these fatty acid species.

Finally, we aimed to assess such changes in the composition of the membrane itself. Using gas chromatography–mass spectrometry of

fatty acid methyl esters (GC–FAME), we detected eight fatty acid species in the membranes of WT, mutant, complementation and overexpression strains grown to mid-exponential phase at 30, 37, and 40 °C (Supplementary Fig. 2c, Supplementary Data 4), and found that temperature affected the obtained composition profiles in a strain-dependent fashion (compositional ANOVA *P* < 0.05, Fig. 4b, Supplementary Fig. 2c). Indeed, they were similar at 30 °C but diverged at higher temperatures (Supplementary Fig. 2d), matching the growth and transcriptome phenotypes observed before (Supplementary Fig. 2a, Fig. 3b). Temperature increase was generally associated with a slight decrease in average acyl chain length, but specifically in the Δ*spv_0647* mutant with an inability to maintain unsaturated fatty acid levels (Fig. 4c), in line with *fabM* downregulation (Fig. 3c). This

**Table 1 | FoldSeek relevant top hits with similar folding to the predicted SPV_0647 structure**

| PDB ID | Bit score | E-value | Probability | Name | Organism | Notes | Reference |
|---|---|---|---|---|---|---|---|
| 4MK6 | 233 | $7.54 \times 10^{-6}$ | 1.00 | DHSK_reg | *Listeria monocytogenes* | Probable dihydroxyacetone (glycerone) kinase regulator | |
| 5NZ1 | 187 | $1.28 \times 10^{-4}$ | 1.00 | EthR | *Mycobacterium tuberculosis* | Bound and allosterically regulated by medium-chain saturated fatty acid hexadecyl octanoate | Frénois et al.[89] |
| 4DW6 | 186 | $1.55 \times 10^{-4}$ | 1.00 | | | | |
| 5F27 | 185 | $4.37 \times 10^{-5}$ | 1.00 | | | | |
| 5NZ0 | 185 | $9.54 \times 10^{-5}$ | 1.00 | | | | |
| 3QOV | 181 | $1.05 \times 10^{-4}$ | 1.00 | | | | |
| 6HO0 | 179 | $1.48 \times 10^{-4}$ | 1.00 | | | | |
| 5NIM | 175 | $2.16 \times 10^{-4}$ | 1.00 | | | | |
| 5MYM | 167 | $2.41 \times 10^{-4}$ | 1.00 | | | | |
| 6HO6 | 161 | $6.40 \times 10^{-4}$ | 1.00 | | | | |
| 2IU5 | 171 | $6.72 \times 10^{-4}$ | 1.00 | DhaS | *Lactococcus lactis* | Dihydroxyacetone kinase activator | Christen et al.[90] |
| 6O6N | 161 | $1.09 \times 10^{-3}$ | 1.00 | FasR | *Mycobacterium tuberculosis* | Activator binds long fatty acyl-CoA | Lara et al.[91] |
| 6O6O | 158 | $1.33 \times 10^{-3}$ | 1.00 | | | | |
| 3LSJ | 156 | $1.54 \times 10^{-3}$ | 1.00 | DesT | *Pseudomonas aeruginosa* | Binds both unsaturated and saturated fatty acids, regulating their ratio | Zhang et al.[92] |
| 6OFO | 155 | $6.40 \times 10^{-4}$ | 1.00 | MtrR | *Neisseria gonorrhoeae* | Regulates hydrophobic substrate efflux, hydrophobic binding pocket | Beggs et al.[93] |
| 5D1W | 153 | $8.57 \times 10^{-4}$ | 1.00 | Rv3249c | *Mycobacterium tuberculosis* | Regulates MmpL-mediated lipid export, binds palmitic acid | Delmar et al.[94] |
| 3WHC | 152 | $1.21 \times 10^{-3}$ | 1.00 | FadR | *Bacillus subtilis* | Binds long-chain acyl-CoAs, shown for stearoyl-CoA | Fujihashi et al.[95] |

All proteins are transcriptional regulators from the TetR family. The full results table is provided in Supplementary Data 3.

phenotype could not only be rescued by *spv_0647* complementation, but also by *fabM* overexpression (Fig. 4b, c, Supplementary Fig. 2c). These results correspond exactly to the growth phenotypes we observed for the same strains (Fig. 4a).

Together, these findings support a model in which SPV_0647 confers heat resistance by modulating the SFA:UFA balance in the membrane through positive regulation of *fabM* and possibly *fakB3*. We have therefore renamed SPV_0647 to FasR, for <u>f</u>atty <u>a</u>cid <u>s</u>aturation <u>r</u>egulator.

**Expression differences do not reflect fitness effects**

We next wanted to know to what extent these and other differential fitness requirements translate to expression patterns. To this effect, we measured both the transcriptome and proteome of *S. pneumoniae* D39V WT grown in six of the infection-mimicking conditions in which CRISPRi-seq was performed (Supplementary Table 1). Using RNA-seq and quantitative, label-free LC–MS, we detected 2146 different RNA species and 870 proteins, of which we retained 2140 and 736, respectively, following standard normalization and imputation methods (Supplementary Data 5). Small, lowly abundant, and membrane proteins were underrepresented in the proteomics measurements (Supplementary Fig. 3a–c). Normalized quantifications of these data were also made accessible in PneumoBrowse 2[29].

In general, transcriptome-proteome correlations resembled those reported before for other organisms[32,35]. Briefly, transcript and protein levels correlated moderately ($R^2 = 0.37$–$0.51$) within growth conditions (Supplementary Fig. 3d), and protein levels tended to increase with transcript levels of individual genes across conditions (Supplementary Fig. 3e). Intuitively, this effect was strongest for genes of which both transcript and protein products were differentially enriched between at least two growth conditions, indicating tighter mRNA-protein regulation (Supplementary Fig. 3f).

Moreover, dimension reduction by multi-omics factor analysis (MOFA)[61] showed reproducible, integrated, condition-specific proteo-transcriptomic landscapes, while accounting for both the paired nature of the data and non-coding transcripts (Fig. 5a). Its first two dimensions (factors) explained most of the variance in either data set (Supplementary Fig. 4a), correlated well with the most highly differentially expressed genes (Fig. 5b), and revealed three main patterns.

Firstly, natural competence was clearly activated in the CSP, FEVER, and CSFMC conditions, distinguishing them from the others (Fig. 5a, c). For these latter two, this can likely be attributed to the relatively high pH in these media (Supplementary Table 1)[62]. The late competence (ComX-regulated) protein response also appeared to be lagged in these two conditions compared to CSP, suggesting slower competence activation (Fig. 5c). Secondly, genes in the *rtg* and *vp1* loci were strongly upregulated in the defined media (Fig. 5c, Supplementary Fig. 4b). These loci both encode a double cell–cell communication loop, where Rgg regulators control glycine–glycine peptide expression[63,64]. It is likely that these quorum-sensing systems were not activated in the complex media due to competition for uptake by the AmiACDEF oligopeptide permease system between the Rgg peptide pheromones and medium-derived peptides[64–66]. Although upregulated, the *rtg* and *vp1* systems were not important for growth in the defined media, further supported by the *amiACDEF* importer genes being essential in the complex media, but not in the defined ones (Figs. 1b and 5d). Lastly, genes responsible for the uptake and metabolism of sucrose and amino sugars were indeed associated with the media containing these types of sugars: C+Y/CSP and LMC/NMC, respectively (Fig. 5c, Supplementary Table 1).

Although the sucrose import operon *scrAK* was upregulated in the corresponding media, our CRISPRi-seq assay indicated it was not essential for growth, which can be explained by the additional presence of glucose (Supplementary Table 1). In contrast, the sucrose catabolism operon *scrBR* was both upregulated and conditionally essential (Fig. 5d). These results imply again the toxicity of phosphotransferase sugar import without degradation, as for GlcNAc. Indeed, recent work from our laboratory showed this toxicity can be alleviated by simultaneous knockdown of *scrA* and *scrB*, inhibiting sucrose import and, with that, likely sugar-phosphate toxicity[60]. These results also imply that, as we showed for GlcNAc, pneumococci import sucrose even in the presence of glucose, at least in sufficient amounts to be toxic in the absence of downstream catabolism.

Strikingly, *nagA*, *nagB*, and *manLMN* were not among the many amino sugar metabolism genes upregulated in NMC and LMC. Instead, these mostly comprised genes involved in uptake (*nanP*, *satABC*) and metabolism (*nanA*, *nanB*, *nanE-1*, *nanK*, *nanE-2*) of other amino sugars not present in the media (Fig. 5d). This observation of orthogonality

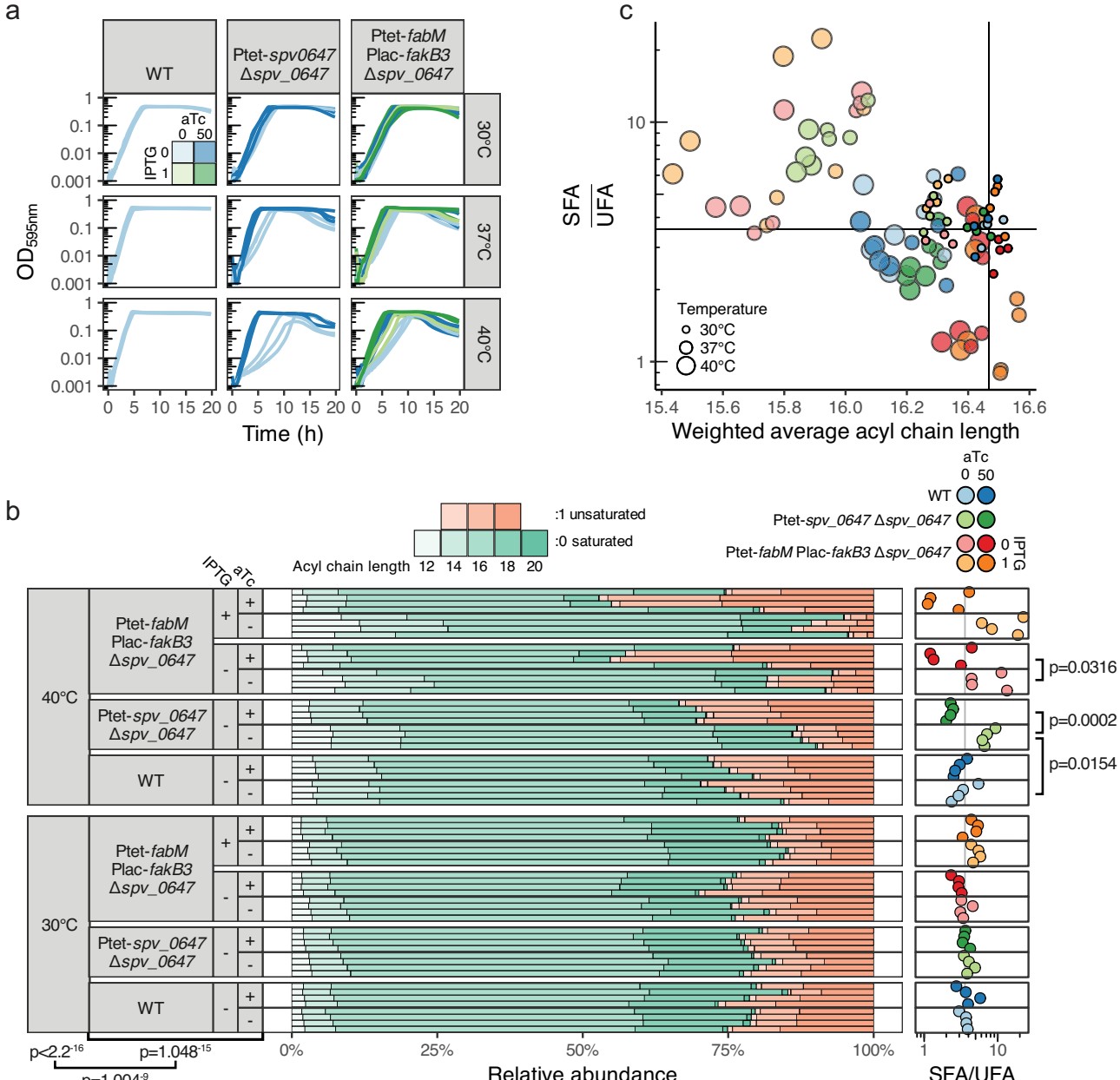

**Fig. 4 | *fasR* (*spv_0647*) affects heat resistance by modulating membrane composition. a** Growth curves of Δ*spv_0647* with either aTc-inducible complementation or aTc- and IPTG-inducible *fabM* and *fakB3* overexpression, respectively, at different temperatures. Biological triplicates are shown. **b** Relative fatty acid composition in membranes of selected strains and temperatures, measured by GC-FAME. The complete set of samples can be found in Supplementary Fig. 2c. *P*-values of temperature and combined strain-induction effects (since inducers have different target genes across strains), and their interaction effect on fatty acid composition for the full sample set are shown below the plot (compositional ANOVA, irl-transformed[88]). Saturated to unsaturated fatty acid ratios (SFA/UFA) were computed from these sample-wise compositions and are shown on the right, including

two-tailed *t*-test *P*-values for relevant contrasts. The gray vertical line represents the average ratio for WT samples without any inducer at 30 °C. Fatty acids that were detected: lauric acid (C12:0), myristic acid (C14:0), palmitic acid (C16:0), stearic acid (C18:0), eicosanoic acid (C20:0), myristoleic acid (C14:1), palmitoleic acid (C16:1), vaccenic acid (C18:1). **c** Average acyl chain length per sample, weighted by relative fatty acid abundance, versus SFA:UFA ratio. Vertical and horizontal lines represent averages for WT grown without inducers at 30 °C. Legend is shared with (**b**). Anhydrotetracycline (aTc) and isopropyl β-ᴅ-1-thiogalactopyranoside (IPTG) were always added to a final concentration of 0 or 50 ng mL⁻¹ and 0 or 1 mM, respectively. Source data are provided as a Source Data file and in Supplementary Data 4.

between gene essentiality and expression, whether referring to the transcriptome or proteome, was dominant across the whole genome and all tested conditions (Fig. 5d, Supplementary Fig. 4c). We did find exceptions to this trend, e.g., the example of *scrB* given above, or the conditional upregulation and essentiality of the HrcA-regulated heat-shock protein-encoding operon at 40 °C, the latter of which was not observed on the proteome level presumably due to the short time

between heat exposure and sample processing (5 min) (Fig. 5d). *fasR* however followed the general trend: despite its importance for growth at 40 °C, it was not differentially expressed, suggesting its basal expression levels are sufficient for this survival phenotype (Fig. 5d). Indeed, our results indicated that differentially expressed genes were rarely differentially essential and vice versa (Supplementary Fig. 4c),

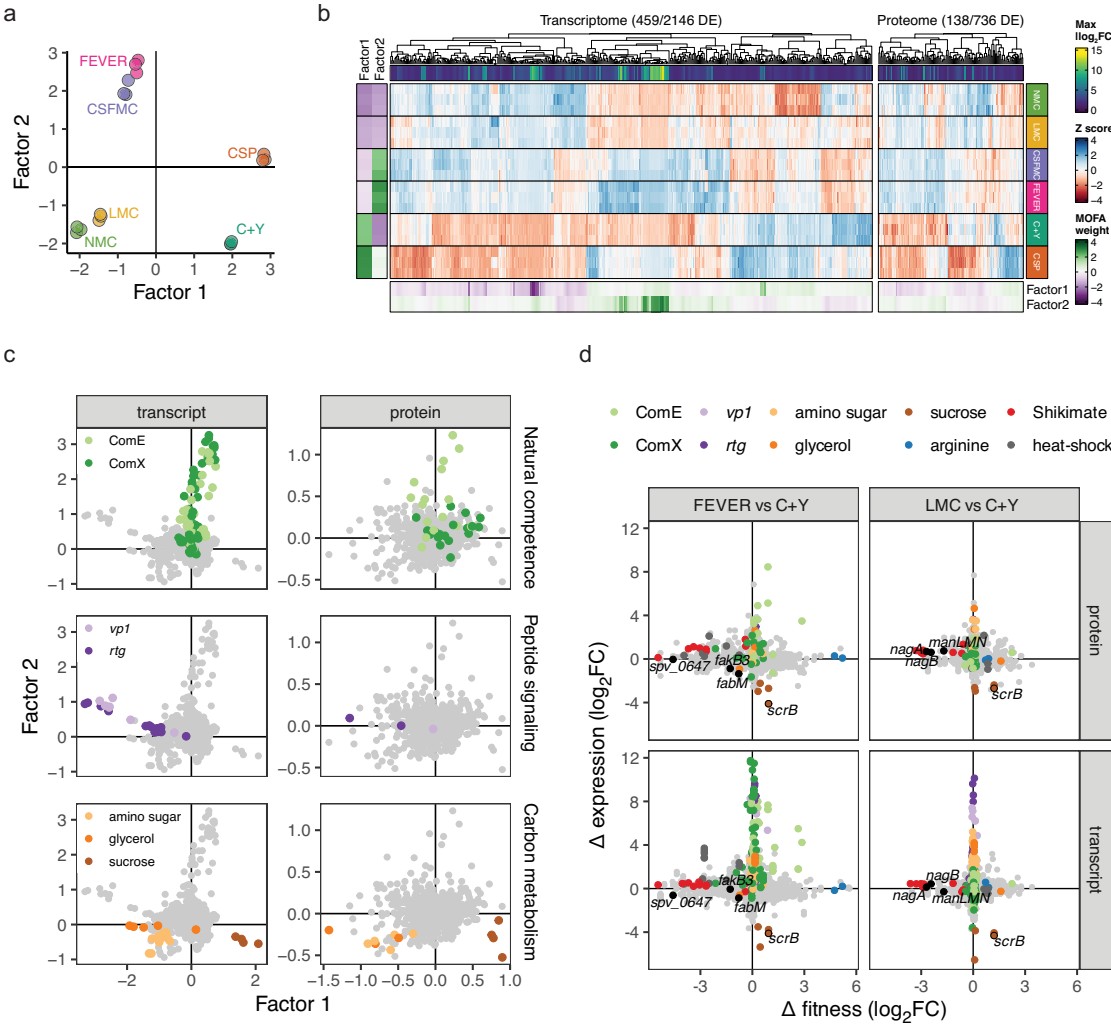

**Fig. 5 | Major differential expression patterns and orthogonality with fitness profiles. a** Dimension reduction by multi-omics factor analysis (MOFA), considering all, including non-coding, transcripts and the paired nature of the proteome-transcriptome samples. **b** MOFA weights correlate well with differential expression patterns between any two growth conditions. **c** MOFA uncovers the main differential expression programs across all growth conditions. Associations of features with samples can be derived from similarity in weight and direction in this space compared to those of the samples in panel (**a**). **d** Orthogonality between differential expression and fitness, illustrated for FEVER and LMC compared to C + Y. Early (ComE) and late (ComX) competence regulons are labeled as defined by Slager and colleagues[45]. *vp1* locus: *mutR1* (*rgg144*), *shp144* (*srf-06*), *vp1* (*spv_0145*), *spv_0146-*

*0148* (*vpoBCD/azlC*)[63]; *rtg* locus: *spv_0112-2103* (*rtgRAXBCHKLPQTUY1W1Z1D2*, *pspA*, *spv_0118*, *spv_2101*)[64]; amino sugar: *nanR* locus (*nanRABE-1PK*, *satABC*, *spv_1490-1492*, *spv_1498*, *spv_1500-1503*, *spv_1505*, *spv_2375*), *spv_1171-nanE-2-spv_1173*, *spv_2320*; glycerol: *glpKOF-spv_2010*, *tpiA*; sucrose: *scrAKBR*; arginine: *argR1*, *ahrC*; Shikimate (as in Fig. 1b): *aroABCDEFGK*, *spv_1151*, *tyrA*, *pheA*; heat-shock (as in Fig. 3a): *ftsH*, *clpP*, *clpE*, *ctsR-clpC*, *groES-groEL*, *hrcA-grpE-dnaK-spv_2171-dnaJ* operons. Differential enrichment was tested using DESeq2 (negative binomial generalized linear model with a Wald test) for transcripts and DEP (linear model with empirical Bayes *t*-statistic moderation) for proteins, with two-tailed tests and *P*-values adjusted for false discovery rate[78,80]. Source data are provided in Supplementary Data 1 and 5.

which fits observations made by others in the same and other organisms[36–41].

## Discussion

In this work, we investigated how pneumococci adapt to different environmental settings in terms of genome-wide gene fitness effects and expression, both on the transcript and protein level. This allowed us to not just re-assess known relationships between these regulatory layers in *S. pneumoniae*, but also to work out specific molecular responses to concrete stimuli.

Global patterns roughly corroborate biology as it is known in other organisms: transcriptomes correlate moderately with proteomes[32–35], and either appear almost entirely statistically independent from genome-wide fitness impact[36–41]. However, Jensen and colleagues noted this does not necessarily imply biological

independence: genes relating to the same general pathways could be either differentially essential or differentially expressed, and as such still coordinated[36]. We see one such example, where *nagA*, *nagB*, and *manLMN* are required for growth on GlcNAc, but not differentially regulated, whereas other amino sugar metabolism genes are, while they are not essential. The data suggest that basal expression levels mostly suffice for bacterial survival in distinct environments, rendering differential expression of essential genes generally redundant. Conversely, differential expression does not signal a changed need for a functional gene copy per se: for instance, a gene can be upregulated in one growth condition compared to another, but essential in both. Moreover, expression-fitness correlations could be masked by genetic redundancy, which could be addressed by multiplexed knockdown approaches such as dual CRISPRi-seq[60,67]. In addition, it is important to acknowledge that the selective pressures that shaped the regulation of

certain genes in response to specific cues in the natural habitat might be absent in our artificial laboratory conditions, whereas those cues can still be present, potentially leading to differential expression without a difference in essentiality. Our data indicate that expression and fitness effects provide almost completely mutually complementary information on how bacteria adapt to different environments. This in turn makes the case for multi-omics approaches to understand how bacteria deal with their surroundings and stresses, including, for instance, antibiotic pressures, with potentially important implications for the treatment of infections. It also suggests that differential expression assays by themselves are not necessarily the best tool to uncover potential therapeutic targets, as they might not point to essentiality.

An environmental factor that is inherently intertwined with infection and disease is ambient temperature. As *S. pneumoniae* moves from the nasopharynx (32 °C) to other parts of the body (37 °C), where it can bring about fever (>38 °C), it faces considerable temperature shifts[68]. This is known to affect the cells in a myriad of ways, including membrane properties such as fluidity and permeability. Bacteria are known to counter these effects through homeoviscous adaptation, i.e., by adjusting the relative levels of different fatty acid types in the membrane[26,27]. Previous studies indicated that such adaptation by *S. pneumoniae* in response to temperature changes was independent of the fatty acid biosynthesis master regulator FabT[19]. Here, we report that *spv_0647* encodes a transcriptional regulator that enables pneumococci to maintain proper saturated:unsaturated fatty acid (SFA:UFA) balance, critical during heat stress, by mediating transcription of *fabM*, and potentially *fakB3* and other genes. As such, we named this gene *fasR*, for fatty acid saturation regulator. In the absence of *fasR*, SFA:UFA ratios increase at higher temperatures, which is conventionally assumed to allow survival at higher temperatures. Notwithstanding, we observed a growth defect, which could specifically be rescued by restoring SFA:UFA balance either by complementation or *fabM* overexpression, presumably increasing UFA biosynthesis substrate levels. Of note, such overexpression did not increase relative UFA levels at 30 °C, implying FabM is normally saturating, in line with previous reports[19]. These results lead us to hypothesize that maintaining SFA:UFA balance, rather than increasing this ratio, confers heat resistance in pneumococci, and that *fasR* and *fabM* play critical roles in that process. Alternatively, fatty acid acyl chain length could also play an important role in maintaining membrane integrity at varying temperatures, which was indeed also affected in the mutant and restored in the complementation and overexpression mutants. Since we did not find putative binding sites for FasR in the promoter regions of differentially expressed genes using motif enrichment analyses with the bioinformatic MEME suite[69], future research may focus on establishing DNA binding sites and sequences using, for example, EMSA and ChIP-seq. Regulation by FasR is likely modulated by a FasR ligand, and although multiple structurally similar proteins have been shown to sense saturated or unsaturated fatty acids by direct binding (Table 1), this remains to be elucidated for FasR. In addition, it could be insightful to examine lipid head groups, which might also influence membrane properties but are missed by GC-FAME, the lipid analysis technique used here[25,26]. Metabolomics techniques could also provide a separate, additional layer of information, as it would be possible to gauge, for instance, the abundance of lipid intermediates, further narrowing down the enzymes potentially regulated by FasR. Moreover, most of the observed differences between the growth conditions tested here were metabolic in nature and could potentially be better understood in the light of metabolomic profiles.

We elaborated on one such instance here, showing that N-acetylglucosamine (GlcNAc) degradation after uptake is essential for pneumococci, and corroborating this is the case for sucrose as well[60]. As both sugars are phosphorylated upon import, we hypothesize these are instances of sugar-phosphate stress[51]. Moreover, these toxicities

are seen in the presence of glucose, suggesting simultaneous uptake of these sugars. Indeed, we showed a slowdown in glucose uptake in the presence of GlcNAc while the growth rate of WT cells was unaffected. Although this challenges the dogma of CcpA-based glucose preference in *S. pneumoniae*[10,13], it may not be illogical from an evolutionary perspective, as glycans such as GlcNAc are far more available for scavenging in its natural niche, the human nasopharynx, than glucose is[6,70,71].

Although we provide a broad overview of the large, systems-level data compendium presented here, we have only worked out a few processes that stood out in detail. Many more biological insights may be concealed in these data, and we encourage the community to use them to their advantage. To facilitate such efforts, we have also integrated our genome-wide fitness, transcript, and protein data sets with our recently renewed genome browser PneumoBrowse 2 (https://veeninglab.com/pneumobrowse)[29]. On top of that, we highlighted some of these avenues to be explored, such as potential sugar-phosphate toxicities or the potential fitness gain upon arginine retention during heat stress. Despite the fact that heat itself has been shown to influence CRISPRi efficiency, we clearly do retrieve the standard core essentialome in our high-temperature growth condition and are therefore confident regarding data quality[72].

Our findings on GlcNAc metabolism and membrane homeostasis contribute to our knowledge of gene function and the biology behind environmental adaptation. We note that many pneumococcal genes remain of unknown function, and that adaptation is a complex phenotype, brought about on multiple, interacting regulatory levels. *S. pneumoniae* is notoriously versatile in terms of niche adaptation, as it can occupy many micro-environments in the human body[1-3]. A deeper understanding of the biology underpinning this capacity, in the pneumococcus as well as other microorganisms, therefore, ultimately also yields a deeper understanding of human health and disease.

## Methods
### Bacterial strains and growth conditions
*Streptococcus pneumoniae* D39V serotype 2 and derivatives were routinely grown at 37 °C on Columbia agar plates with 2.5–5% (v/v) defibrinated sheep blood (CBA, Thermo Scientific) at 5% $CO_2$, or in sealed 5 mL culture tubes with C + Y liquid medium (pH 6.8) without shaking, supplemented with 0.5 µg mL$^{-1}$ erythromycin, 0.5 µg mL$^{-1}$ tetracycline, or 50 ng mL$^{-1}$ anhydrotetracycline when appropriate. Liquid cultures were routinely inoculated at 100× dilutions from pre-cultures cultivated from frozen, isogenic stock cultures (16% glycerol). Strains used in this study are listed in Supplementary Table 2. Other growth conditions and media were prepared as described by Aprianto and colleagues (2018)[31], with the only adaptation of continuous culturing at 40 °C in the FEVER condition for the CRISPRi-seq assay. Growth assays for sugar preference were performed in a C + Y liquid or agar plate background without added sugars, supplemented with 9.4 mM glucose, 9.4 or 0.94 mM N-acetyl-D-glucosamine (Sigma-Aldrich, A3286), or both sugars as appropriate.

### Mutant strain construction
Donor DNA constructs carrying the insert of interest with -1000 bp flanking regions homologous to the insertion site were produced with a one-pot Golden Gate assembly strategy using Type II restriction enzymes BsaI, Esp3I, or SapI (New England Biolabs)[73]. Restriction sites were introduced via the primers during PCR amplification. Used oligonucleotides and restriction enzymes are listed in Supplementary Table 3. We used the PT5-3 variant of the P$_{tet}$ promoter as characterized by Sorg and colleagues (2020)[74].

Subsequent transformation with the donor DNA was performed as described previously[44]. Briefly, pneumococci were cultured at 37 °C to the early exponential phase (OD595 ~ 0.1) followed by the addition of 0.1 µg mL competence-stimulating peptide 1 (CSP-1) and growth for another 12 min to activate competence. 100 µL activated culture was

mixed with donor DNA at a concentration of 1 ng µL and further cultured at 30 °C for 20 min. 900 µL fresh C + Y was added, and the culture was cultivated at 37 °C for another 1.5 h for the transformants to recover. The culture was plated and incubated overnight as described above. Colonies were re-streaked on plates and again incubated overnight. Transformant colonies were picked and grown in liquid C + Y until OD595 ~ 0.3, and stocked in 14−20% glycerol. Genotypes of mutant strains were confirmed by Sanger sequencing (Microsynth).

## Growth curves

Pre-cultures were grown in C + Y to OD595 ~ 0.1 at 37 °C, diluted 100× in the appropriate medium, and loaded into flat-bottom 96-well plates at 250 µL per well. 50 ng mL$^{-1}$ anhydrotetracycline or 1 mM IPTG was added to both pre-cultures and dilutions where appropriate. Experiments were always performed in triplicate (three separate pre-cultures), unless specifically stated in the figure. For each replicate, a representative curve was chosen out of three technical replicates (wells within the same plate), on the basis of the most frequent appearance between the other two technical replicates across all time points. Blanks were added to each plate to assess potential contamination. In experiments where glucose levels were also measured, the number of technical replicates equaled the number of glucose measurement time points to allow subsampling for that purpose. Optical density was measured every 10 min at 595 nm in a plate reader (Tecan MPlex, F200 or M200 series). In temperature variation experiments, three identical plates were prepared and measured simultaneously in three different plate readers, each set at a different temperature (30, 37, or 40 °C). Plates were sealed with parafilm in this case to avoid excessive evaporation. Raw OD values were normalized per well to the theoretical start OD of 0.001 by subtraction, and lower values were also set to this theoretical minimum.

## Glucose oxidation assays

Samples were obtained by pausing the plate reader during growth curve assays at time points indicated in the figures, transferring the contents of technical replicates (200 µL) to microtubes, and resuming OD measurements as fast as possible. Microtubes were immediately spun down on a tabletop mini centrifuge for 3 min to pellet the cells, after which 150 µL supernatant was transferred to a new microtube. The first time point sample (0 h) was taken directly from the growth curve pre-culture and treated in the same way. Samples were snap-frozen using liquid nitrogen and stored at −80 °C until glucose measurements.

Glucose concentrations were measured with a Glucose (GO) Assay Kit (Sigma-Aldrich GAGO20) according to the instructions of the manufacturer, except for the total sample volumes. We scaled all volumes down so that samples were 150 µL instead of 5 mL, allowing for higher-throughput measurements using flat-bottom 96-well plates in a Tecan plate reader, instead of cuvettes.

## Proteo-transcriptomics

Wild-type cells were pre-cultured in each respective growth medium to OD600 ~ 0.1 and diluted to OD600 ~ 0.05. CSP cultures were supplemented with 0.1 µg mL$^{-1}$ CSP-1 for 20 min, and FEVER cultures were transferred to 40 °C for 5 min. Each of three replicates was split into two subsamples, one of which was subjected to RNA-seq and the other to quantitative LC−MS.

Total RNA was extracted and cDNA libraries were constructed as before, without rRNA depletion[31]. Libraries were sequenced on an Illumina NextSeq machine at GeneCore, EMBL Heidelberg. Read quality was checked with FastQC (v0.11.5, https://www.bioinformatics. babraham.ac.uk/projects/fastqc/) before and after trimming off TruSeq3 adapters, leading and trailing bases below a phred score of 3, cutting regions if average phred scores went below 20 in a sliding window of 5 bases, and only keeping reads with a minimum length of

50 bases using Trimmomatic (v0.36)[75]. Reads were aligned to the *S. pneumoniae* D39V reference genome (CP027540) using STAR (v2.5.3a)[76] and reverse strand transcripts were counted with feature-Counts (Subread v1.5.3)[77], using multi-mapping, overlap, and fractional count modes, as before[31]. Downstream analyses were done with DESeq2 (v1.34.0)[78] in R (v4.1.1), where differential expression was tested against an absolute log$_2$ fold change of 1 at an alpha of 0.05, and counts were normalized for Principal Component Analysis with a blind rlog transformation[78]. Transcripts per million (TPM) were calculated per sample with $n$ genes for each gene $i$ as: $\text{TPM}_i = \frac{c_i \cdot 10^6}{l_i \cdot \sum_{j=1}^{n} \frac{c_j}{l_j}}$, where $c$ and $l$ represent the raw transcript counts and lengths, respectively.

Cells in the proteomics subsamples were lysed with a bead beater and further treated for LC−MS at the Proteomics−Mass Spectrometry Service Facility, University of Groningen. Briefly, sample volumes were reduced by freeze drying, protein concentrations were determined with a BCA assay (Thermo, 23252), and samples were reconstituted in a 100 mM ammonium bicarbonate buffer. Alkylation of 100 µg protein was achieved by adding iodoacetamide to a final concentration of 40 mM and incubation for 45 min at room temperature, in the dark. Samples were diluted 2× in 100 mM ammonium bicarbonate, and overnight digestion was performed at 37 °C, 400 rpm with mass spectrometry grade trypsin (Promega, V5280) using a 1:50 trypsin:protein (µg:µg) ratio. The reaction was stopped by adding trifluoroacetic acid to a final concentration of 1%. Pierce® C18 tips (Thermo, 87784) were used for sample cleanup by solid phase extraction according to the manufacturer's instructions. The elute fraction was dried under vacuum and reconstituted with 20 µL 2% acetonitrile and 0.1% formic acid (FA). Peptide separation was performed with 2 µL peptide sample using a nano-flow chromatography system (Thermo, EASY nLC II) equipped with a reversed phase HPLC column (75 µm, 15 cm) packed in-house with C18 resin (Dr. Maisch, ReproSil-Pur C18−AQ, 3 µm resin) using a linear gradient from 95% solvent A (0.1% FA, 2% acetonitrile) and 5% solvent B (99.9% acetonitrile, 0.1% FA) to 28% solvent B over 90 min at a flow rate of 200 nL min$^{-1}$. The total MS time was 120 min. The peptide and peptide fragment masses were determined by an electrospray ionization mass spectrometer (Thermo, LTQ-Orbi-trap XL).

Peptides were mapped to the *S. pneumoniae* D39V (CP027540) protein fasta file and quantified as both label-free quantification (LFQ) and intensity Based Absolute Quantification (iBAQ) values with MaxQuant[79], using a false discovery rate (FDR) cutoff of 0.01. Downstream analyses were performed with R package DEP (v1.16.0)[80] using LFQ values as input, where a protein was considered differentially enriched if its absolute log$_2$-fold change was significantly >1, with an FDR-adjusted *P*-value below 0.05. We retained only proteins that were detected in at least two out of three replicates per condition, normalized with a variance-stabilizing transformation, and imputed missing values with the MinProb method as implemented in DEP, since values were not missing at random, but biased towards lower intensities (Supplementary Fig. 3). Gene Ontology (GO) term enrichment analysis was carried out using R package clusterProfiler (4.2.2)[81]. For visualization purposes, we normalized for library size bias per sample as proteins per million (PrPM): $\text{PrPM}_i = \frac{\text{iBAQ}_i \cdot 10^6}{\sum_{j=1}^{n} \text{iBAQ}_j}$, akin to TPM as described above.

We used the blind rlog and variance-stabilized quantifications of transcripts and proteins with non-zero variance across samples as input for Multi-Omics Factor Analysis using the R package MOFA2 (v1.4.0)[61] with default settings.

## RNA-seq Δ*spv_0647*

Wild-type (VL1) and Δ*spv_0647* (VL6297) strains (Supplementary Table 2) were pre-cultured in C + Y medium at 30 and 37 °C until OD595 ~ 0.3. Pre-cultures were diluted to OD595 ~ 0.01 in

quadruplicates and grown at 30 °C and 37 °C until OD595 ~ 0.3–0.4. Cells were pelleted by centrifugation (4 °C, 10,000×*g*, 5 min), supernatants removed, and pellets were stored at −80 °C after snap-freezing with liquid nitrogen.

Total RNA was isolated with a High Pure RNA Isolation Kit (Roche, 11828665001) as before, with minor adaptations[31]. Briefly, the pellets were resuspended in 400 μL Tris–EDTA buffer and transferred to tubes with 50 μL SDS 10%, 500 μL phenol–CHCl₃, and glass beads. Cells were lysed with a bead beater (3× 45 s with 45 s breaks) and pelleted by centrifugation (4 °C, 21,000×*g*, 15 min). 300 μL of the aqueous phase was mixed into 400 μL lysis/binding buffer. The samples were loaded onto columns and centrifuged (8000×*g*, 30 s). 100 μL DNase mix (90 μL buffer, 10 μL DNase I) was loaded onto the column filter and incubated for 1 h at room temperature. Samples were washed once with wash buffer I (500 μL) and twice with wash buffer II (first time 500 μL, second time 200 μL) by centrifugation (8000×*g*, 30 s). Samples were eluted in 50 μL elution buffer and incubated for 10 min at room temperature. Sample quality was checked by NanoDrop and Fraction Analyzer (Agilent Technologies), and samples were stored at −80 °C. Samples had RNA Quality Numbers (RQN) between 5.7 and 7.8.

Per strain-temperature combination, the three samples with minimal potential gDNA contamination and the highest RQN were selected for cDNA library preparation and sequencing at the Genomic Technologies Facility, University of Lausanne. Briefly, RNA-seq libraries were prepared from 100 ng of total RNA with the Illumina Stranded mRNA Prep reagents (Illumina) using a unique dual indexing strategy and following the official protocols. The polyA selection step was replaced by an rRNA depletion step with RiboCop for Bacteria, mixed bacterial samples, and reagents (Lexogen). Libraries were quantified by a fluorometric method (QubIT, Life Technologies), and their quality was assessed on a Fragment Analyzer (Agilent Technologies). Sequencing was performed on an Illumina NovaSeq 6000 for 100 cycles, single read. Sequencing data were demultiplexed using the bcl2fastq2 Conversion Software (v2.20, Illumina).

Read quality was checked with FastQC (v0.11.9) and MultiQC (v1.15)[82] before and after trimming off leading and trailing bases below a phred score of 3, cutting regions if average phred scores went below 20 in a sliding window of 5 bases, and only keeping reads with a minimum length of 50 bases using Trimmomatic (v0.36)[75]. Reads were aligned to the *S. pneumoniae* D39V reference genome (CP027540) using bowtie2 (v2.4.5)[83], using soft-clipping (with the "--local" option) to account for any remaining partial adapters.

Transcript counts were extracted with featureCounts (v2.0.6)[77] and downstream analyses were performed with DEseq2[78] in R, as described for the proteo-transcriptome experiment.

## CRISPRi-seq

An IPTG-inducible genome-wide CRISPRi library was used as described before[43,44]. Briefly, the library was pre-cultured from frozen stock (16% glycerol in the respective media, OD595 = 0.1) by 100× dilution in the respective growth conditions (Supplementary Table 1) to OD595 ~ 0.1, followed by 100× dilution supplemented with 1 mM IPTG for CRISPRi induction and CSP-1 (0.1 μg mL⁻¹) when appropriate, to yield quadruplicates for each growth condition with and without IPTG. Samples were grown to OD595 ~ 0.1 once (LMC, NMC, CSFMC, FEVER; 4–5 mL culture in 5 mL culture tubes) or twice (C + Y, CSP, THY, BMC; 10 mL in 50 mL conical tubes) by 100× back-dilution, corresponding to 7–14 generations of exponential, sample-wide growth (Supplementary Data 1). Pellets were harvested on ice, centrifuging once (7 generations) or twice (14 generations) (4 °C, 15 min at 4000×*g* and 5 min at 12,000×*g*, respectively), discarding supernatant and resuspending in PBS. gDNA isolation, library preparation, and sequencing were done as described in our published protocols[44]. CRISPRi-induced NMC samples 54 and 55 got mixed up during gDNA isolation and were henceforth regarded as technical replicates.

Samples were split over two sequencing runs on an Illumina MiniSeq system with our published custom sequencing protocol[44]. sgRNA counts were extracted with 2FAST2Q (v2.5.2)[84] using default settings (minimal phred score 30, one mismatch allowed). Differential fitness analyses were performed with DESeq2 (v1.34.0)[78] in R (v4.1.1), testing against a minimal (difference in) absolute log₂-fold change of 1 at an alpha of 0.05 for statistical significance. We collapsed the induced NMC technical replicates into one biological replicate with the DESeq2 function "collapseReplicates()".

Given growth at a rate of $2^n$, fitness quantifications on a log₂FC should scale linearly across conditions. We observed this was not the case between conditions grown for a different sample-wide number of generations, implying a strong library composition bias and rendering DESeq2-based interaction effects uninformative. We corrected this bias by estimating generation numbers per CRISPRi strain in each induced sample, linearizing them over all samples with a LOESS transformation, and recomputing corresponding sgRNA counts. These were rounded and used as DESeq2 input. Sample-wide CRISPRi induction generation numbers (0, 7, or 14) were scaled and centered with the built-in R function scale(), and together with growth condition and their interaction term were used as explanatory variables in the DESeq2 design formula. So, default DESeq2 methods were applied after we normalized for the non-linear generation effect.

Specifically, for each of $k$ sgRNAs in a sample, the raw count $c$ was first normalized to $c^*$ to correct for library size bias: $c^*_{t1} = \log_2(1 + \frac{10^6 \cdot c}{\sum_{i=1}^{k} c_i})$, where $t1$ indicates this concerns counts after growth (as measured). We used uninduced sample counts as a proxy for the relative starting distribution of CRISPRi strains per sample prior to induction, i.e., at $t0$. Given bacterial growth occurs at a rate of $2^m$, so that in general $c_{t1} = c_{t0} \cdot 2^m$, the relative counts per strain at $t0$ are here: $c_{t0} = \frac{c^*_{t1}}{2^m}$, where in our case $m$ is 7 or 14. To obtain a more robust estimate, we averaged $c_{t0}$ per strain across each set of four uninduced replicate samples, yielding $\bar{c}_{t0}$. This represents an estimate of the relative starting counts per CRISPRi strain, or sgRNA, for all samples within a given growth condition. Following the same standard growth equation as above, the relative counts after growth should then equal $c^*_{t1} = \bar{c}_{t0} \cdot 2^n$, and so we estimated the strain-wise generation numbers in each induced sample as: $n = \log_2(\frac{0.5 + c^*_{t1}}{\bar{c}_{t0}})$, where a pseudocount of 0.5 was added to avoid zeroes. Together, this yields the generation number matrix $\mathbf{N}$, where rows represent CRISPRi strains, and columns represent induced samples of all growth conditions. Generation number estimates were then linearized with the cyclicloess method of the normalizeBetweenArrays() function from the limma R package (v3.50.1)[85], with $\mathbf{N}$ as input and otherwise default parameters. The resulting matrix columns represent the corrected generation number estimates $n^*$ for each strain per induced sample, which we then used to estimate relative strain counts after growth per induced sample following the same standard growth dynamics described above: $\hat{c}_{t1} = \hat{c}_{t0} \cdot 2^{n^*}$, where $\hat{c}_{t0} = \frac{c}{2^m}$. Lastly, we scaled these corrected counts to their original library size as: $\hat{c}^*_{t1} = \hat{c}_{t1} \cdot \frac{\sum_{i=1}^{k} c_i}{\sum_{i=1}^{k} \hat{c}_{t1_i}}$. These normalized sgRNA counts for the induced samples were subsequently used as DESeq2 input for downstream analyses, together with the original sgRNA counts of the uninduced samples.

## Foldseek

Foldseek[59] was accessed through the online submission portal (https://search.foldseek.com) and ran with UniProt accession number A0A0H2ZQ31, for *spv_0647*. Only matches with crystalized protein structures from the PDB100 database were considered for this work.

## Dual CRISPRi-seq analysis

Fitness scores (log$_2$FC values) for every unique pair of 869 sgRNAs were obtained from Dénéréaz and colleagues[60]. Scores of sgRNAs in combination with themselves (twice the same sgRNA in the same CRISPRi strain) were used as baseline fitness estimates of the corresponding sgRNA, as in the original study.

## GC-FAME

Strains were pre-cultured at 30, 37, or 40 °C with or without 50 ng mL$^{-1}$ anhydrotetracycline and 1 mM IPTG, as appropriate, to an OD595 ~ 0.3 and concentrated 10× by centrifugation (3 min, 8000×$g$). Quadruplicate samples were grown from these pre-cultures in the same growth conditions, as appropriate, in volumes of 45 mL per sample to OD595 ~ 0.2–0.3. Samples were kept at their respective growth temperatures throughout the experiment. Cells were pelleted by centrifugation (15 min, 1968×$g$), resuspended in ~1 mL of the remaining medium after discarding supernatant, transferred to 2 mL screw-cap tubes, and centrifuged again (5 min, 20,238×$g$). Cells were washed by supernatant removal, resuspension in 1 mL PBS, and centrifugation (5 min, 20,238×$g$). Supernatant was removed again, and pellets were stored at −80 °C following snap-freezing with liquid nitrogen.

For derivatization of fatty acid methyl esters, frozen cell pellets were resuspended and then vortexed in 100 μL concentrated sulfuric acid (96%) and 200 μL methanol. Samples were then boiled by incubation in boiling water for 5 min and allowed to cool to RT. 300 μL of dichloromethane was added and the samples were then vortexed and centrifuged 1 min at 15,000×$g$. The organic (bottom) layer was transferred to a new tube containing a pinch of Na$_2$SO$_4$ (to remove any remaining water), mixed by vortexing, and centrifuged 1 min at 15,000×$g$. The supernatant was then transferred to an HPLC tube and stored at 4 °C until being loaded on the GC−MS. 1 μL of sample was injected into an Agilent 7890 Gas Chromatograph equipped with an Agilent G3903-63011 column, an Agilent 5977A Mass Detector, and an Agilent 7693 Autoinjector. The carrier gas was helium, and the column oven temperature program was the following: 150 °C for 0.5 min; ramp temperature 25 °C/min to 230 °C, then hold for 1 min; ramp temperature 5 °C/min to 245 °C, then hold for 1 min. Spectra from 50 to 500 $m/z$ were collected after a 4 min solvent delay. The ion source temperature was 230 °C.

Raw data were exported as CDF files, which were filtered to remove empty scans and converted to .mzML files using MZmine 4 (mzio.io)[86]. Spectral alignment, deconvolution, and relative abundance analyses were performed using MS-Hub as part of the global natural products social molecular networking (GNPS) GC−MS EI Data Analysis pipeline[87]. Peaks representing methylated fatty acids were identified based on comparison of spectra and retention times to those obtained using TraceCERT 37 component fatty acid methyl ester standard (Sigma Aldrich) across the same instrumentation protocol. For each sample, peak intensities were summed per unique fatty acid. These values were standardized to sum to 100 per sample, and analyzed with the R package compositions (v2.0.8) to account for the particular biases and inherent complexities of compositional data[88]. Specifically, we used the acomp() function to transform the data for principal component analysis, and the irl transformation in combination with a compositional (mlm) ANOVA for hypothesis testing.

## Reporting summary

Further information on research design is available in the Nature Portfolio Reporting Summary linked to this article.

## Data availability

For the proteo-transcriptomic profiling, raw RNA-seq data are available on SRA, accession number PRJNA527271, and LC−MS data on the UCSD MassIVE repository, accession number MSV000097932. Raw CRISPRi-seq data are available on SRA, accession number PRJNA1262882. RNA-seq data for the mutant experiment can be found on SRA, accession number PRJNA1262992, and GC-FAME data on the UCSD MassIVE repository, accession number MSV000097619. Source data are provided with this paper.

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

## Acknowledgements

We would like to thank Andrew Quinn for advice on the use of glucose oxidation assays, Florian Bock for practical biochemistry guidance, Johann Mignolet and Julien Dénéréaz for providing genetic constructs, Axel Janssen for integrating our data with PneumoBrowse 2, Kathrin Fröhlich for bringing the phenomenon of sugar-phosphate stress to our attention, and James Sáenz for valuable insights regarding membrane properties. Moreover, we thank Rieza Aprianto for proteo-transcriptomic sample preparation, and Johan Hekelaar at the Proteomics Facility of the University of Groningen for the LC–MS quantifications and his help in

proteomics analyses. RNA-seq for the proteo-transcriptome experiment was performed at GeneCore, EMBL, Heidelberg. Library preparation and RNA-seq of the *spv_0647* mutant and WT strain were performed at the Lausanne Genomic Technologies Facility, University of Lausanne, Switzerland. This work was supported by Swiss National Science Foundation (SNSF) PostDoc Mobility fellowship P500PB_225439 (V.d.B.), ERC consolidator grant 771534, SNSF grants 310030_192517, 310030_200792, NCCR 'AntiResist' 51NF40_180541 (J.W.V.), and NIH/NIDCR R00-029228 (J.L.B.).

## Author contributions

V.d.B. performed experiments, analyzed data, and wrote the manuscript. X.L. supervised experiments. J.T. and M.B. performed experiments and analyzed data. J.L.B. supervised experiments, analyzed data, and edited the manuscript. J.W.V. supervised the study and edited the manuscript.

## Competing interests

J.W.V. is a scientific advisory board member at i-Seq Biotechnology. The remaining authors declare no competing interests.
