## [Transparent Peer Review file · Nature Communications]

Multi-omics profiling reveals atypical sugar utilization and a key membrane composition regulator in *Streptococcus pneumoniae*

Corresponding Author: Professor Jan-Willem Veening

Version 0:

Reviewer comments:

Reviewer #1

(Remarks to the Author)

The manuscript by de Bakker and colleagues, titled "Multi-omics profiling reveals atypical sugar utilization and identifies a key membrane composition regulator in *Streptococcus pneumoniae*" describes how diverse in vitro infection-related conditions influence the functional genomics of *S. pneumoniae*. The authors use transcriptomics, proteomics, and CRISPRi-based mutagenesis to map global changes. The primary aim of the study (as perceived by me) is to ascertain the alignment of these diverse strategies. The study shows that there is a significant overlap between RNA and protein regulation in response to the diverse conditions. However, the alignment with the mutagenesis screen is poor when studied at the gene level, with the exception of a select few candidates such as *scrB*. Although the authors suggest that there is a greater potential for consistency when analyzed as pathways, this has not been explored systematically. The color coding in the figures appears to be via manual annotation, instead of a fully unbiased approach. I would recommend that the authors explore this opportunity. Further, this may have to be performed as "differentially expressed genes or differentially represented mutants", and not "up vs down", as both can occur when analyzing entire pathways.

The manuscript itself is structured such that the authors first explore the CRISPRi analyses independently to describe a phenomenon related to GlcNAc uptake and catabolism, and then regarding membrane modulation following heat stress, prior to comparative multi-omics. I understand why the authors structure the manuscript accordingly, as limited interconnectivity across these three aspects was observed, and instead they delve deeper into the facets where strong phenotypes were observed. But it does yield a somewhat disjointed manuscript. In regard to the figures on the omics (e.g. Fig 1), the authors have been able to generate very dense figures, that allow for very complete omics data sets to be shown, but it limits discerning the differences within pathways. For example, Fig 1b could be moved to the supplementary, as the pathways are not clearly visible. Instead, I would recommend reconstructing the pathways with individual gene/protein candidates shown more clearly, as it will be more informative to the reader. Although Fig 2e serves this purpose, there is no integration of the omics data, and this could be expanded on. Similarly, fig 4b and d are highly informative, but close to indigestible to readers less familiar with the field / broader audience (ie Nat Comms). The manuscript text makes half-sentence references to these graphs and are therefore probably suitable for relocation to the sup mat.

Overall, this is a large body of work, with arguably the most important message being that genome-wide mutagenesis studies compliment transcriptome and proteome studies, consistent with previous studies (in other bacteria). The findings that demonstrate that glucose may not always be the preferred carbon source are significant. This would be interesting to study in an animal infection model. The identification of the novel fatty acid saturation regulator is an important part of understanding lipid modulation and host adaptation. This also has possible implications beyond pneumococci.

The manuscript is quite succinct, which is justifiable when discussing the global omics approaches, but the two independent molecular findings would benefit from more in-depth omics data integration and description. The independent mutant studies and complementation for each of the two molecular mechanisms are excellent, thereby providing solid evidence of the systems.

Specific comments:

Lines 38-40: The first part of the final sentence of the abstract seems more appropriate for the start of the results section of

the abstract.

Introduction (detailed background - Lines 53-63): Too succinct, no info on *ccpA*, lipid/fatty acid homeostasis and other multi condition Tn-seq works in pneumococci.

Extended Data table 1: This table should include more detail on the media compositions (base media, C and N-source, pH!). I know the same media were used in the NAR paper, but it would really support the readability of this work. The growth conditions (start OD (from ON culture [media type], or plate?), volume, CO₂, final OD, time to OD, vessel type) for the RNA seq/proteomics and CRISPRi should also be denoted. Are these all identical? This is important when trying to compare the output data.

Figure 1: define all abbreviations in legend. Fig 1b is near superfluous as the pathway genes can't be seen anyway.

Line 135: The glc vs GlcNAc competition assays should also be performed in minimal media where no other carbon sources are present.

Fig 2a: The [Glc] symbols in the panels are overlapping, and this doesn't allow for the visualization of the replicates. If the top one is white (or marginally transparent), then I wouldn't be able to see the "true greenness" of those below. Although I appreciate the elegance of the data presentation, it could create concerns, especially without any statistical tests, which should be included.

Fig 2e and Line 393: This is highly speculative without any intracellular metabolomics, please revise the text in the Fig legend to be less verbose. Also, denote genes/proteins for Glc uptake other than ManLMN.

Fig S1D: Remove 3rd / bottom panel as this is the same as Fig 2b.

Line 164: Amend to "...growth in the same CSFMC media, but at 37C."

Line 164: The time delay induced by heat stress should be mentioned, it's very dramatic.

Fig 3a: Are *fakB3* and *fakM* any different in the CRISPRi-seq analyses under heat stress? Please discuss.

Lines 208-211: this needs clarification.

Discussion: What are the FasR substrates, could it be UFA abundances directly?

Lines 255-260: Instead of all the overly complicated graphs, can you just show a simple UFA/SFA graph with statistical tests?

Fig 5c and d: please use the same color for the same "pathway".

Line 404: The link isn't active.

Line 428: The use of C+Y when changing carbon sources is surprising.

Line 448: Was there CO₂ enrichment in the growth curve analyses?

Reviewer #2

(Remarks to the Author)

This is an excellent report from de Bakker, et al, that advances the Veening Lab's ongoing systems-level analysis of *Streptococcus pneumoniae*'s gene and protein regulatory networks. In a prior work, transcriptomic datasets described differential gene expression programs during several infection-mimicking conditions. Here, CRISPRi-seq supplements transcriptomic and proteomic analysis to provide fitness values for all genes under 22 different, host-mimicking (and classic laboratory) conditions. The data are sure to be invaluable to the field and a major strength of the project is the infrastructure built by the Veening lab to provide an open-access, web-based, genome-level data repository called PneumoBrowse 2. Impressively, findings of the current work are displayed in multi-dimensional figures that would impress Edward Tufte. The report digs deeper into two areas and reveals novel findings. First, the concept of carbon-source priority is challenged by findings that GlcNAc is consumed simultaneously alongside glucose and mutants of GlcNAc catabolism that cause sugar-phosphate accumulation are toxic. Secondly, FEVER conditions led to identifying a regulator of fatty acid saturation. It is highly likely that many additional discoveries await in the rigorous and validated datasets that have been generated here. Overall, this was a pleasure to read and I find the work to be of exceptional quality.

The only aspect of the work I found to be non-intuitive was the finding that higher temperatures led to an increase in unsaturated fatty acids. I would have predicted that at higher temperatures the bacterium would want incorporate more saturated FAs, not unsaturated, to prevent the membrane from becoming overly fluid and thus likely leaky. My interpretation of the results in Figure 4 is that induced expression of *fabM* rescues mutants of *fasR* (*spv_0647*), and that increased levels of unsaturated FAs were seen. Could the authors indicate if this was an unpredicted result, and if so, why might higher

expression of FabM benefit growth in higher temperatures?

Minor suggestion: consider re-writing the statement at lines 104-106. I realize this is intended to serve as a segue into the next section, but I found it to be confusing as written.

Reviewer #3

(Remarks to the Author)

The manuscript by de Bakker et al uses large scale screening of a CRISPR-i library in a variety of different growth conditions in the *S pneumoniae* strain D39 to explore the concept of differential essentiality. In the context of this work they are adding to their already incredibly useful database of differential gene expression. This alone will add significantly to the advancement of research on pneumococcus but the authors go on to note several examples of how this data set helps expand our understanding of pneumococcal biology. For example, they show convincingly that *S pneumoniae* takes up GlcNAc, even in the presence of glucose and that GlcNAc intermediates can be toxic without the function of *nagA* and *nagB*. They additionally make the observation that arginine levels are likely protective under high temperature conditions and that a previously unknown protein, *spv_0647*, appears to encode a regulator that is required for altering the membrane fatty acid content that allows cells to survive under elevated (ie fever associated) temperature conditions. The work goes on to compare the transcriptomics to proteomics under a variety of relevant conditions and make some interesting observations with this approach. The work is impactful, clearly described and the conclusions are supported by the data presented. Figures are excellent and easy to follow. This was a pleasure to read. I only have a few suggestions on wording that may help the reader.

1. Lines 100-102, it may help the reader to explain the data here in more detail, ie the shikimate pathway showed fitness reductions only in defined media while the *ami* genes showed the opposite.
2. Line 188, it would be helpful to define "dose dependent" since the inducer, glucose and GlcNAc are all being added (ie dose dependent on the concentration of GlyNAc).
3. Line 134-145/ It may be helpful to explicitly spell out the hypothesis here, that glucose and GlcNAc compete for the same *manLMN* importer and when both sugars are present, both are imported.
4. Line 152-153 – "imply the presence of toxic intermediates" may be clearer wording.
5. Line 162-3 is confusing, all the assessed conditions are consistently grown under the specified condition. I think the authors are getting at the fact that the high temperature condition and the CSF condition only differ by the temperature of incubation? Can this be rephrased for clarity?
6. Line 167 – It would be helpful to label the arginine genes in Figure 1b as was done for *ami*/ shikimate genes.
7. Lines 304-308 (and 405-406) – The hypothesis here (307-308) is not clear. It is known that Rgg regulators rely on *Ami* import of their SHP signals and that complex media interferes with peptide import. This is thought to be due to competition between the peptide pheromones with media derived peptides for the importer. It seems clear based on what is already known that Rgg regulatory pathways would be expected to be upregulated in defined media without excess peptides. The authors should include the existing data in this regard into their discussion of this finding.

We would like to thank the referees for their thorough and insightful reviews, and we are grateful for their appreciation of our work. We have considered and processed all the input they provided, including updates to all figures, which has in our opinion resulted in a clearer and more rounded manuscript. Below, we address every comment point by point.

Reviewer #1 (Remarks to the Author):

The manuscript by de Bakker and colleagues, titled "Multi-omics profiling reveals atypical sugar utilization and identifies a key membrane composition regulator in *Streptococcus pneumoniae*" describes how diverse in vitro infection-related conditions influence the functional genomics of *S. pneumoniae*. The authors use transcriptomics, proteomics, and CRISPRi-based mutagenesis to map global changes. The primary aim of the study (as perceived by me) is to ascertain the alignment of these diverse strategies. The study shows that there is a significant overlap between RNA and protein regulation in response to the diverse conditions. However, the alignment with the mutagenesis screen is poor when studied at the gene level, with the exception of a select few candidates such as *scrB*. Although the authors suggest that there is a greater potential for consistency when analyzed as pathways, this has not been explored systematically. The color coding in the figures appears to be via manual annotation, instead of a fully unbiased approach. I would recommend that the authors explore this opportunity. Further, this may have to be performed as "differentially expressed genes or differentially represented mutants", and not "up vs down", as both can occur when analyzing entire pathways.

We thank the reviewer for their thorough reading of the manuscript, and their thoughtful comments. Given any of the various genome-wide assays we have done, we reasoned there are two general ways to approach the data: (1) start with existing pathway annotations and test for their differential enrichment between conditions (e.g., GO gene set enrichment analysis), or (2) start by identifying the most strongly (in terms of \log_2FC) and most certainly (in terms of adjusted p-value) differentially enriched genes/transcript/proteins/sgRNAs, and determine whether they already are, or otherwise can manually be, assigned to specific pathways or processes using multiple data bases, existing literature and tools, including UniProt, FoldSeek, KEGG, the EMBL homology e-mapper EGGNOG and PaperBlast. For this particular manuscript, we opted for the second strategy, for various reasons: (a) Our main interest was in describing the strongest (\log_2FC) and most certain (P_{adj}) global patterns in the data; (b) In doing so, we identified whether these genes are part of specific pathways or processes, and as such still described and labeled these pathways and processes, but with

a focus on those that are truly differentially enriched (up, down or both); (c) Existing annotation data bases are often incomplete, updated infrequently, and while informative, biased towards known biology. By adopting a manually curated approach, including cross-referencing with multiple data bases and tools such as those listed above, we ensured that we did not miss major patterns and that we also inspected individual features for which gene set annotations are missing. For instance, the signaling loci *vp1* and *rtg* generally do not have GO terms associated with them and *spv_1151* links the Shikimate pathway to specific amino acid biosynthesis, which we only found due to manual searches and cross-referencing with other data bases such as KEGG. We believe that in this way, we have covered the major patterns in the data, while maintaining interpretability of found hits and pathways and their biological meaning. The reviewer's comment did inspire us to return to the gene annotations and labels in each of our figures, and we realized they (1) were not always consistent between figures, (2) did not clearly distinguish between direct hits and polar effects and (3) were often not clearly stated in the legends. We have corrected these issues by re-making all figures, adapting color coding and labeling and adjusting the legends. This has made the figures throughout the manuscript easier to compare and increased transparency, so we are grateful the reviewer raised questions regarding annotations and pointed us to these inconsistencies and ambiguities.

The manuscript itself is structured such that the authors first explore the CRISPRi analyses independently to describe a phenomenon related to GlcNAc uptake and catabolism, and then regarding membrane modulation following heat stress, prior to comparative multi-omics. I understand why the authors structure the manuscript accordingly, as limited interconnectivity across these three aspects was observed, and instead they delve deeper into the facets where strong phenotypes were observed. But it does yield a somewhat disjointed manuscript. In regard to the figures on the omics (e.g. Fig 1), the authors have been able to generate very dense figures, that allow for very complete omics data sets to be shown, but it limits discerning the differences within pathways. For example, Fig 1b could be moved to the supplementary, as the pathways are not clearly visible. Instead, I would recommend reconstructing the pathways with individual gene/protein candidates shown more clearly, as it will be more informative to the reader. Although Fig 2e serves this purpose, there is no integration of the omics data, and this could be expanded on. Similarly, fig 4b and d are highly informative, but close to indigestible to readers less familiar with the field / broader audience (ie Nat Comms). The manuscript text makes half-sentence references to these graphs and are therefore probably suitable for relocation to the sup mat.

We are glad the reviewer appreciates our efforts to comprehensively visualize the complex data sets and we acknowledge that patterns on the pathway, or even gene level are hard to discern in this fashion. We have taken steps to address the concerns.

Even though most of these pathways, such as the Shikimate pathway, are not studied in more detail in this project, we agree that the reader should be able to assess the effects we describe more clearly in the relevant figure. For the same reason, we do find it important to keep a version of Fig. 1b in the main text: so that the reader may examine our data and the hits we discuss in a broader context, e.g., how many sgRNAs were differentially enriched, how do they relate across conditions, or what do these data “look like”? So, instead of moving this panel to the supplementary materials, we expanded it to address the reviewer’s valid point: we added a sub-panel with the hits we discuss in the context of their reconstructed metabolic pathways and linked this panel to the corresponding locations in the original heatmap. This includes the hits shown in Fig. 2e, for which the reviewer pointed out that the CRISPRi-seq data was not clearly depicted. In addition, we inverted the color scale of the minimal absolute fitness differences between complex and defined media to make more subtle differences discernable for the reader. We also adapted the main text referring to this panel to reflect the biological function of the pathways we discuss, to guide the broad readership of *Nature Communications* through the results. Lastly, the renewed Fig. 1 now also includes the arginine metabolism hits discussed later in the manuscript, as requested by another reviewer.

This last modification also aides in rounding the manuscript story, making it less disjoint. To expand on that effort, we have also adapted the main text in the Results section on expression-fitness comparisons to discuss the differentially expressed signaling pathways in the context of AmiACDEF essentiality, standardized pathway annotations across all figures and added the main CRISPRi-seq hits we discuss to the expression-fitness graph in Fig. 5d. This will facilitate cross-figure and -omics comparisons. New pathway annotations now either exclude polar effects, or explicitly describe them (e.g., in figure or graph legends), to avoid confusion with true pathway members. All members are now also always mentioned in corresponding figure legends.

As the reviewer points out, the original Fig. 4 contained exhaustive, but also redundant information. Specifically, the PCA (original Fig. 4b) showed much overlapping information compared to the original Fig. 4d, but in a more abstract fashion. We therefore moved it to the supplementary materials (Extended Data Fig. 2d). We instead added the *spv_0647* knockout/complementation mutant to the former Fig. 4c (new panel 4b) so that readers may still assess its profile in the absence of the PCA. We also adjusted the renewed figures to address other remarks made by the reviewer below and enhanced ease of interpretation by (1) adding more white space to Fig. 4b to emphasize contrasts of interest, (2) add a simple SFA:UFA plot to Fig. 4b with corresponding statistical tests for relevant contrasts, (3) have the color coding correspond to Fig. 4c so that the only added information there is

represented by point size (temperature) and x-axis (chain length) and (4) adjust Fig. 4c y-axis to facilitate log scale interpretation.

Overall, this is a large body of work, with arguably the most important message being that genome-wide mutagenesis studies compliment transcriptome and proteome studies, consistent with previous studies (in other bacteria). The findings that demonstrate that glucose may not always be the preferred carbon source are significant. This would be interesting to study in an animal infection model. The identification of the novel fatty acid saturation regulator is an important part of understanding lipid modulation and host adaptation. This also has possible implications beyond pneumococci.

The manuscript is quite succinct, which is justifiable when discussing the global omics approaches, but the two independent molecular findings would benefit from more in-depth omics data integration and description. The independent mutant studies and complementation for each of the two molecular mechanisms are excellent, thereby providing solid evidence of the systems.

Thank you for your appreciation. We have adapted both the main text and Fig. 5d to include more information on the multi-omics profiles of the two main molecular systems that we studied. We believe this also aids in rounding the manuscript.

Specific comments:

Lines 38-40: The first part of the final sentence of the abstract seems more appropriate for the start of the results section of the abstract.

We agree and have changed this sentence in the abstract.

Introduction (detailed background - Lines 53-63): Too succinct, no info on ccpA, lipid/fatty acid homeostasis and other multi condition Tn-seq works in pneumococci.

Done. The Introduction has been extensively modified to give broad background information on the two main processes studied in more detail in this paper: (1) carbohydrate metabolism with a focus on NagAB, CcpA/CCR and ManLMN and (2) fatty acid metabolism and membrane homeostasis. We also included a reference to an extensive Tn-seq multi-condition study. We agree this information was missing previously and believe the renewed version will aid the reader in understanding the work more readily.

Extended Data table 1: This table should include more detail on the media compositions (base media, C and N-source, pH!). I know the same media were used in the NAR paper, but it would really support the readability of this work. The growth conditions (start OD (from ON culture [media type], or plate?), volume, CO₂, final OD, time to OD, vessel type) for the RNA

seq/proteomics and CRISPRi should also be denoted. Are these all identical? This is important when trying to compare the output data.

We added relevant details and all information that deviated from the standard growth conditions as described at the start of the Methods section to the appropriate Methods sections and Extended Data Table 1.

Figure 1: define all abbreviations in legend. Fig 1b is near superfluous as the pathway genes can't be seen anyway.

Done; Fig. 1b has been drastically modified (see comment above).

Line 135: The glc vs GlcNAc competition assays should also be performed in minimal media where no other carbon sources are present.

Please see our response to the comment for line 428 below.

Fig 2a: The [Glc] symbols in the panels are overlapping, and this doesn't allow for the visualization of the replicates. If the top one is white (or marginally transparent), then I wouldn't be able to see the "true greenness" of those below. Although I appreciate the elegance of the data presentation, it could create concerns, especially without any statistical tests, which should be included.

Thank you for drawing our attention to this matter, we care greatly about proper visualization and demonstrating reproducibility. As such, we changed the figure to display glucose concentration as point size instead of by color. Since the points are no longer filled, readers can now appreciate all replicate measurements as point edges, or rings, where overlapping points can still be distinguished. We also added appropriate statistical test outcomes for contrasts of interest and indicated these in the figure.

Fig 2e and Line 393: This is highly speculative without any intracellular metabolomics, please revise the text in the Fig legend to be less verbose. Also, denote genes/proteins for Glc uptake other than ManLMN.

Although it is known that there must be alternative glucose import systems in *S. pneumoniae* because *manLMN* has been shown to do this, but knockout strains can nonetheless grow fine on glucose, their identity remains elusive. We have adapted the figure legend to include this information along with the appropriate references. In addition, we have changed the text to explicitly reflect the strictly hypothetical nature of our model in panel e.

Fig S1D: Remove 3rd / bottom panel as this is the same as Fig 2b.

Done.

Line 164: Amend to “...growth in the same CSFMC media, but at 37C.”

We acknowledge this should be phrased more straight to the point. Since the medium is formally called CSFLM, for CSF-like medium, but we wanted to avoid introducing more terms than necessary in the main text for clarity, we rephrased it to “...we compared it with growth in CSFMC: the same medium, but at 37C.” This should solve the issue in a similar way.

Line 164: The time delay induced by heat stress should be mentioned, it's very dramatic.

Done.

Fig 3a: Are fakB3 and fakM any different in the CRISPRi-seq analyses under heat stress? Please discuss.

We now added these gene labels to the plot (Fig. 3a) to show they are not differentially essential in this assay. We also added labels to Fig. 5d and text to the Results section to discuss this result and place it in the context of the story.

Lines 208-211: this needs clarification.

Done.

Discussion: What are the FasR substrates, could it be UFA abundances directly?

Yes, this is definitely an option. In fact, the proteins we present in Table 1 do not only fold similarly to FasR, but many of them have been shown to directly bind specific fatty acids. The reviewer correctly identifies working out the sensing and regulatory mechanism as a major outstanding question, and accordingly we have added text to this effect in the discussion section.

Lines 255-260: Instead of all the overly complicated graphs, can you just show a simple UFA/SFA graph with statistical tests?

We acknowledge that this figure contains plots with redundant information and should be simplified for interpretability. We have therefore moved the PCA to the Extended Data section, simplified plots that remained, and added a panel with SFA:UFA ratios and corresponding statistical test outcomes for relevant contrasts, as requested. We chose to keep the relevant bar graphs to allow the reader to get an understanding of what these data look like and to assess reproducibility in the absence of the more abstract PCA. We also kept a simplified version of the panel showing SFA:UFA ratio versus acyl chain length, as these represent relevant, interpretable phenotypes calculated from the more complex compositions. This panel includes all temperature groups, enabling the reader to gauge their effects without the need to show them in the bar graphs and again eliminating the need for a

PCA in this figure. We feel that with these changes, we have addressed this valid concern and that the figure has become more digestible and clearer.

Fig 5c and d: please use the same color for the same “pathway”.

Done.

Line 404: The link isn't active.

We have checked this in multiple internet browsers, but it is working for us. Please report back if the issue persists. For now, we will assume that it was a matter of bad timing, as the server has had a few brief hiccups in the past, which should now be resolved.

Line 428: The use of C+Y when changing carbon sources is surprising.

While we understand the reviewer's surprise, the reasoning was as follows: we first wanted to check if we could reproduce *nagA* and *nagB* essentiality with complementary methods in another medium, to strengthen our belief that this phenotype was GlcNAc-dependent. Since we could see in our CRISPRi-seq data that *nagA* and *nagB* were not essential in C+Y (now clearly illustrated in the renewed Fig. 1b following the reviewer's helpful suggestions above), our standard laboratory medium, we decided to check if we could *change* these genes' statuses from dispensable to essential simply by replacing the sugars present even in this medium. This worked, but unexpectedly also rendered the genes essential when growing the knockout strains on both glucose and GlcNAc, which is where the story then proceeds. We previously skipped over the reasoning that led us to use C+Y in these experiments (to match and compare with the CRISPRi-seq conditions) and can see how it can be a surprising finding in the methods section. We therefore added a line of context in the relevant Results section.

Line 448: Was there CO₂ enrichment in the growth curve analyses?

Thank you for pointing this out, it was indeed not clearly phrased. No, the cultures for the growth curves followed our routine liquid cultivation method in sealed culture tubes, and not in the incubator explicitly controlling CO₂ levels. This is now rephrased at the start of the Methods section under “Bacterial strains and growth conditions”.

Reviewer #2 (Remarks to the Author):

This is an excellent report from de Bakker, et al, that advances the Veening Lab's ongoing systems-level analysis of *Streptococcus pneumoniae*'s gene and protein regulatory

networks. In a prior work, transcriptomic datasets described differential gene expression programs during several infection-mimicking conditions. Here, CRISPRi-seq supplements transcriptomic and proteomic analysis to provide fitness values for all genes under 22 different, host-mimicking (and classic laboratory) conditions. The data are sure to be invaluable to the field and a major strength of the project is the infrastructure built by the Veening lab to provide an open-access, web-based, genome-level data repository called PneumoBrowse 2. Impressively, findings of the current work are displayed in multi-dimensional figures that would impress Edward Tufte. The report digs deeper into two areas and reveals novel findings. First, the concept of carbon-source priority is challenged by findings that GlcNAc is consumed simultaneously alongside glucose and mutants of GlcNAc catabolism that cause sugar-phosphate accumulation are toxic. Secondly, FEVER conditions led to identifying a regulator of fatty acid saturation. It is highly likely that many additional discoveries await in the rigorous and validated datasets that have been generated here. Overall, this was a pleasure to read and I find the work to be of exceptional quality.

We are very grateful for the reviewer's appreciation, and particularly the Edward Tufte reference, as we spent considerable time on creating comprehensive but interpretable visualizations of the vast amount of data we amassed, and take pride in the result.

The only aspect of the work I found to be non-intuitive was the finding that higher temperatures led to an increase in unsaturated fatty acids. I would have predicted that at higher temperatures the bacterium would want incorporate more saturated FAs, not unsaturated, to prevent the membrane from becoming overly fluid and thus likely leaky. My interpretation of the results in Figure 4 is that induced expression of *fabM* rescues mutants of *fasR* (*spv_0647*), and that increased levels of unsaturated FAs were seen. Could the authors indicate if this was an unpredicted result, and if so, why might higher expression of *FabM* benefit growth in higher temperatures?

The reviewer is correct in that this was indeed an unexpected result. However, we can see from our data that even the WT strain did not incorporate more SFA in its membrane at higher temperatures but rather maintained a stable SFA:UFA ratio (original Fig. 4c, renewed Fig. 4b). This implies that pneumococci do not necessarily respond to an increase in temperature by increasing relative SFA levels in the membrane, as would be expected based on the traditional model of homeoviscous adaptation. The *fasR* mutant fails to uphold this SFA:UFA ratio at higher temperatures, either producing less UFA or more SFA, and we observe a growth defect. We therefore believe that deleting *fasR* renders the pneumococci incapable of maintaining a stable SFA:UFA ratio at higher temperatures, rather than inhibiting its increase.

Specifically, overexpression of *fabM* should produce more UFA biosynthesis substrate, generating the *potential* for increased UFA production to compensate for the SFA:UFA increase in the mutant. Of note, *fabM* overexpression did not decrease SFA:UFA ratio at 30C, implying that FabM is normally saturating, as has been shown by others. In other words, we arrive at a model where pneumococci strive to maintain stable SFA:UFA balance across temperatures, *fasR* is critical to achieve that goal through regulation of *fabM* and potentially other genes and *fabM* overexpression can restore that balance if the ratio becomes too high.

We appreciated the reviewer's question and realized this piece of interpretation was missing from the Discussion section. We have added it. We have also added some discussion on alternative explanations: the SFA:UFA ratio is not the only factor influencing membrane homeostasis or even fluidity specifically. For instance, polar head group variability and acyl chain length may play an important role as well.

Minor suggestion: consider re-writing the statement at lines 104-106. I realize this is intended to serve as a segue into the next section, but I found it to be confusing as written.

Done.

Reviewer #3 (Remarks to the Author):

The manuscript by de Bakker et al uses large scale screening of a CRISPR-i library in a variety of different growth conditions in the *S pneumoniae* strain D39 to explore the concept of differential essentiality. In the context of this work they are adding to their already incredibly useful database of differential gene expression. This alone will add significantly to the advancement of research on pneumococcus but the authors go on to note several examples of how this data set helps expand our understanding of pneumococcal biology. For example, they show convincingly that *S pneumoniae* takes up GlcNAC, even in the presence of glucose and that GlcNAc intermediates can be toxic without the function of *nagA* and *nagB*. They additionally make the observation that arginine levels are likely protective under high temperature conditions and that a previously unknown protein, *spv_0647*, appears to encode a regulator that is required for altering the membrane fatty acid content that allows cells to survive under elevated (ie fever associated) temperature conditions. The work goes on to compare the transcriptomics to proteomics under a variety of relevant conditions and make some interesting observations with this approach. The work is impactful, clearly described and the conclusions are supported by the data presented. Figures are excellent

and easy to follow. This was a pleasure to read. I only have a few suggestions on wording that may help the reader.

Thank you for your kind words.

1. Lines 100-102, it may help the reader to explain the data here in more detail, ie the shikimate pathway showed fitness reductions only in defined media while the ami genes showed the opposite.

Done.

2. Line 188, it would be helpful to define “dose dependent” since the inducer, glucose and GlcNAc are all being added (ie dose dependent on the concentration of GlyNAc).

Done.

3. Line 134-145/ It may be helpful to explicitly spell out the hypothesis here, that glucose and GlcNAc compete for the same manLMN importer and when both sugars are present, both are imported.

Done, although we did not name ManLMN here specifically, as it is only introduced in the following paragraph.

4. Line 152-153 – “imply the presence of toxic intermediates” may be clearer wording.

Done.

5. Line 162-3 is confusing, all the assessed conditions are consistently grown under the specified condition. I think the authors are getting at the fact that the high temperature condition and the CSF condition only differ by the temperature of incubation? Can this be rephrased for clarity?

Done.

6. Line 167 – It would be helpful to label the arginine genes in Figure 1b as was done for ami/shikimate genes.

Done. We also expanded this figure following the suggestions of another reviewer, making the specific gene fitness profiles discernable to further facilitate interpretation and comparisons across figures.

7. Lines 304-308 (and 405-406) – The hypothesis here (307-308) is not clear. It is known that Rgg regulators rely on Ami import of their SHP signals and that complex media interferes with peptide import. This is thought to be due to competition between the peptide pheromones with media derived peptides for the importer. It seems clear based on what is already known

that Rgg regulatory pathways would be expected to be upregulated in defined media without excess peptides. The authors should include the existing data in this regard into their discussion of this finding.

Thank you for drawing our attention to this point. We have adapted the text at the lines indicated by the reviewer and added context and references. In addition, we have coupled this to our renewed Fig. 1b, which now shows our essentiality data on the Ami importer genes more clearly.